# miR-103 promotes endothelial maladaptation by targeting lncWDR59

Lucia Natarelli[1], Claudia Geißler[1], Gergely Csaba[2], Yuanyuan Wei[1], Mengyu Zhu[1], Andrea di Francesco[1,3], Petra Hartmann[1], Ralf Zimmer [2] & Andreas Schober [1,4,5]

Blood flow at arterial bifurcations and curvatures is naturally disturbed. Endothelial cells (ECs) fail to adapt to disturbed flow, which transcriptionally direct ECs toward a maladapted phenotype, characterized by chronic regeneration of injured ECs. MicroRNAs (miRNAs) can regulate EC maladaptation through targeting of protein-coding RNAs. However, long non-coding RNAs (lncRNAs), known epigenetic regulators of biological processes, can also be miRNA targets, but their contribution on EC maladaptation is unclear. Here we show that hyperlipidemia- and oxLDL-induced upregulation of miR-103 inhibits EC proliferation and promotes endothelial DNA damage through targeting of novel lncWDR59. MiR-103 impedes lncWDR59 interaction with Notch1-inhibitor Numb, therefore affecting Notch1-induced EC proliferation. Moreover, miR-103 increases the susceptibility of proliferating ECs to oxLDL-induced mitotic aberrations, characterized by an increased micronucleic formation and DNA damage accumulation, by affecting Notch1-related β-catenin co-activation. Collectively, these data indicate that miR-103 programs ECs toward a maladapted phenotype through targeting of lncWDR59, which may promote atherosclerosis.

[1] Experimental Vascular Medicine, Institute for Cardiovascular Prevention, Ludwig-Maximilians University Munich, Pettenkoferstrasse 9, 80336 Munich, Germany. [2] Institute for Informatics, Ludwig-Maximilians University Munich, Oettingenstraße 67, 80538 Munich, Germany. [3] Department of Cardiac, Thoracic and Vascular Sciences, University of Padova, Via Giustiniani 2, 35128 Padova, Italy. [4] Institute for Molecular Cardiovascular Research, RWTH Aachen University, Pauwelsstrasse 30, 52074 Aachen, Germany. [5] DZHK (German Centre for Cardiovascular Research), partner site Munich Heart Alliance, Biedersteiner Strasse 29, 80802 Munich, Germany. Correspondence and requests for materials should be addressed to A.S. (email: aschober@med.lmu.de)

Arterial endothelial cells (ECs) are exposed to different types of blood flow patterns and shear stresses, which determine the endothelial phenotype. Indeed, laminar flow results in high shear stress and favors a quiescent EC phenotype characterized by a low proliferation rate and a reduced inflammatory activation[1,2]. In contrast, disturbed flow at arterial branching points and curvatures constantly induces low-grade injury to ECs resulting in apoptosis and inflammation[3–5]. Moreover, the injury triggers a repair process by EC proliferation, which is essential to maintain the endothelial lining[5,6]; however, because disturbed flow-induced EC injury is physiologic, endothelial repair remains ineffective[6,7]. Hence, EC maladaptation generates the Achilles heel of the arterial system, vulnerable to additional damages by hyperlipidemia, which inhibits EC proliferation[8,9] and sustains chronic endothelial wound healing[9,10].

Different types of shear stresses regulate the expression of specific microRNAs (miRNAs), which play key roles in EC injury and proliferation by negatively regulating post-transcriptional gene expression[11]. Indeed, endothelial Dicer affects inflammation and promotes atherosclerosis partly by miR-103-mediated suppression of Krüppel-like factor-4 (Klf4)[12], which control the phenotypic adaptation of ECs to mechanical stress. In addition, miRNAs also affects EC proliferation. Indeed, miR-126-5p decreases atherosclerosis and improve endothelial proliferation by targeting Notch1-inhibitor Dlk1[6]. MiRNAs can also interact with long non-coding RNAs (lncRNAs), which have a broad spectrum of action as epigenetic regulators[13,14] and scaffold molecules[15], to promote angiogenesis and cell differentiation. Among these, lincRNA-p21[16], ANRIL[15] and MALAT1[16] emerged as miRNA decoys during atherosclerosis. However, the role of lncRNAs as targets of miRNA on EC maladaptation is still unclear.

We found that downregulation of miRNAs after Dicer knockout in ECs increases the expression of lncRNA transcripts, including the novel lncRNA WD Repeated Domain 59 (lncWDR59). In particular, targeting of lncWDR59 by miR-103 affects EC proliferation due to reduced Notch1 activation. Moreover, Notch1 activity on EC proliferation is limited by lncWDR59 through β-catenin, which prevents oxidized low-density lipoprotein (oxLDL)-induced micronucleic DNA damage accumulation in proliferating cells. Hence, miRNAs might promote EC maladaptation by negatively regulating lncRNA transcripts, which can reprogram ECs toward a more adaptive phenotype.

## Results

**Dicer regulates lncRNA expression in ECs during atherosclerosis.** After a 12-week high fat diet (HFD) feeding period, the expression of 172 out of 2926 lncRNAs was differentially regulated in atherosclerotic arteries from EC-Dicer$^{flox}$ mice compared with EC-Dicer$^{WT}$ mice (fold change (FC) $\geq$ 1.5, $P < 0.05$, $n = 2$ mice per group) as determined by genome-wide microarray analysis (Fig. 1a). Among the differentially regulated lncRNAs, the expression of 97 and of 75 lncRNAs was increased and decreased, respectively, in EC-Dicer$^{flox}$ mice (Fig. 1a). Among the 97 upregulated lncRNAs, 6 were downregulated while 5 were also upregulated in the atherosclerotic arteries from Apoe$^{-/-}$ mice after Dicer knockout in myeloid cells (Supplementary Table 2). For 39 upregulated lncRNAs, including Jpx, Dlx1as, and Six3os, which play a role in cell viability[17] and differentiation[18], the transcript sequence was previously determined (Supplementary Table 2). To determine the sequences of the other 58 upregulated lncRNAs, including the novel most significantly upregulated lncRNA lncWDR59, we performed a strand-specific RNA

sequencing (RNA-seq) of the RNA isolated from murine aortic ECs (MAoECs). In addition to 15,199 protein-coding genes (fragments per kilobase million (FPKM) > 1), like the endothelial-specific genes Cdh5, Nos3, and Pecam1 (Supplementary Figure 1A, B), we found that 34,524 lncRNA genes, including 18,478 annotated lncRNAs such as Malat1, Neat1, and Fendrr, and 16,046 novel lncRNA genes were expressed in MAoECs (Supplementary Figure 1A,B). All 39 annotated lncRNAs upregulated in EC-Dicer$^{flox}$ mice were also expressed in MAoECs (Supplementary Table 2). In addition, read fragments of 5 out of the 58 non-annotated lncRNAs upregulated in EC-Dicer$^{flox}$ mice, including lncWDR59, were identified by de novo transcript assembly combined with chromatin-state maps[19,20] (Supplementary Figure 1C–E and Fig. 1c). Among these five lncRNAs, the sequence of one new lncRNA, here called Leonardo, was fully determined by RNA-seq (Fig. 1c, Supplementary Figure 1D and Supplementary Notes). Leonardo gene is 1.317 kb long, consists of 5 exons and 4 introns, and is located between Gm29666 and nucleoporin 50 (Nup50) genes on the negative strand of chromosome 15. Analysis of the read fragments showed that the Leonardo transcript is spliced into a 792 bp-long mature lncRNA, which is predicted to fold into a thermodynamically stable secondary structure (Fig. 1c and Supplementary Figure 2). In contrast to Leonardo, the sequence of lncWDR59 was only partially determined by RNA-seq. To discover the lncWDR59 full sequence, we performed a modified RACE-PCR (rapid amplification of cDNA ends–polymerase chain reaction) (Supplementary Figure 1E and Supplementary Notes). The lncWDR59 transcript was 1.61 kb long, derived from one exon located on chromosome 8 between the fatty acid 2-hydroxylase (fa2h) and the wdr59 genes (Fig. 1c and Supplementary Figure 1E). Similarly as Leonardo, lncWDR59 transcript showed a thermodynamically stable secondary structure with characteristic functional lncRNA loops (Supplementary Figure 2 and Supplementary Table 3). Moreover, Coding Protein Calculator analysis showed a low protein-coding ability for lncWDR59, similar to Malat1 (used as positive control).

**Let-7b and miR-103 target lncRNAs.** Dicer knockout in ECs reduces the expression of 20 miRNAs, including miR-103 and let-7b, in atherosclerotic arteries[12]. To study whether miRNAs target lncRNAs in ECs during atherosclerosis, the sequences of 44 lncRNAs upregulated in EC-Dicer$^{flox}$ mice were analyzed for putative binding sites (BSs) for the 20 miRNAs downregulated after Dicer knockout. A total of 189 BSs were predicted in 36 lncRNAs (Fig. 1d and Supplementary Figure 3). The highest number of BSs was found for miR-103 (22 BSs in 21 lncRNAs) and let-7b (22 BSs in 16 lncRNAs) in the sequence of 28 lncRNAs, including 9 lncRNAs that contain BSs for both miRNAs (Fig. 1d and Supplementary Figure 3). These data suggest that miR-103 and let-7b play a prominent role in targeting endothelial lncRNAs. The expression of 15 out of 28 lncRNAs with a number of high-quantity BSs for miR-103 and let-7b was studied in EC-Dicer$^{flox}$ mice and EC-Dicer$^{WT}$ by quantitative PCR (qPCR). Upregulation of 10 lncRNAs, including lncWDR59, Leonardo, and lnc039159, in atherosclerotic arteries from EC-Dicer knockout mice was confirmed by qPCR ($n = 4$–7 mice per group). By contrast, the other 5 lncRNAs and Malat1, which does not contain miR-103 or let-7b BSs, were not differentially regulated (Fig. 1e). To study whether loss of miR-103 and let-7b mediates the upregulation of the 10 lncRNAs in ECs, MAoECs were treated with Locked Nucleic Acid (LNA)-modified inhibitors of miR-103 or let-7b. Treatment with let-7b inhibitors increased Leonardo, Jpx, lnc064083, lnc039159, and lnc002669, whereas lnc003783 and lnc052749 were not affected (Fig. 1f).

Moreover, inhibition of miR-103 upregulated *lncWDR59*, *Gm4258*, *Jpx*, *lnc064083* and *lnc039159*, but not *lnc002669* and *lnc073657* (Fig. 1f).

To assess whether let-7b and miR-103 target lncRNAs in the RISC (RNA-induced silencing complex) of ECs, GW182 immunoprecipitation was performed in MAoECs treated with let-7b or miR-103 mimics. Treatment with let-7b mimics enriched the transcripts of Leonardo, lnc002669, and lnc039159

in the RISC, but not those of Jpx and lnc064083 (Fig. 1g). MiR-103 mimics enriched lncWDR59, Gm4258, lnc064083, and Jpx transcripts in the RISC (Fig. 1g). These results indicate a prominent role of miR-103 in the targeting of lncRNAs in ECs.

**miR-103 targets lncWDR59.** Next, the effect of pro-atherogenic conditions on the expression of the miR-103 targets lncWDR59,

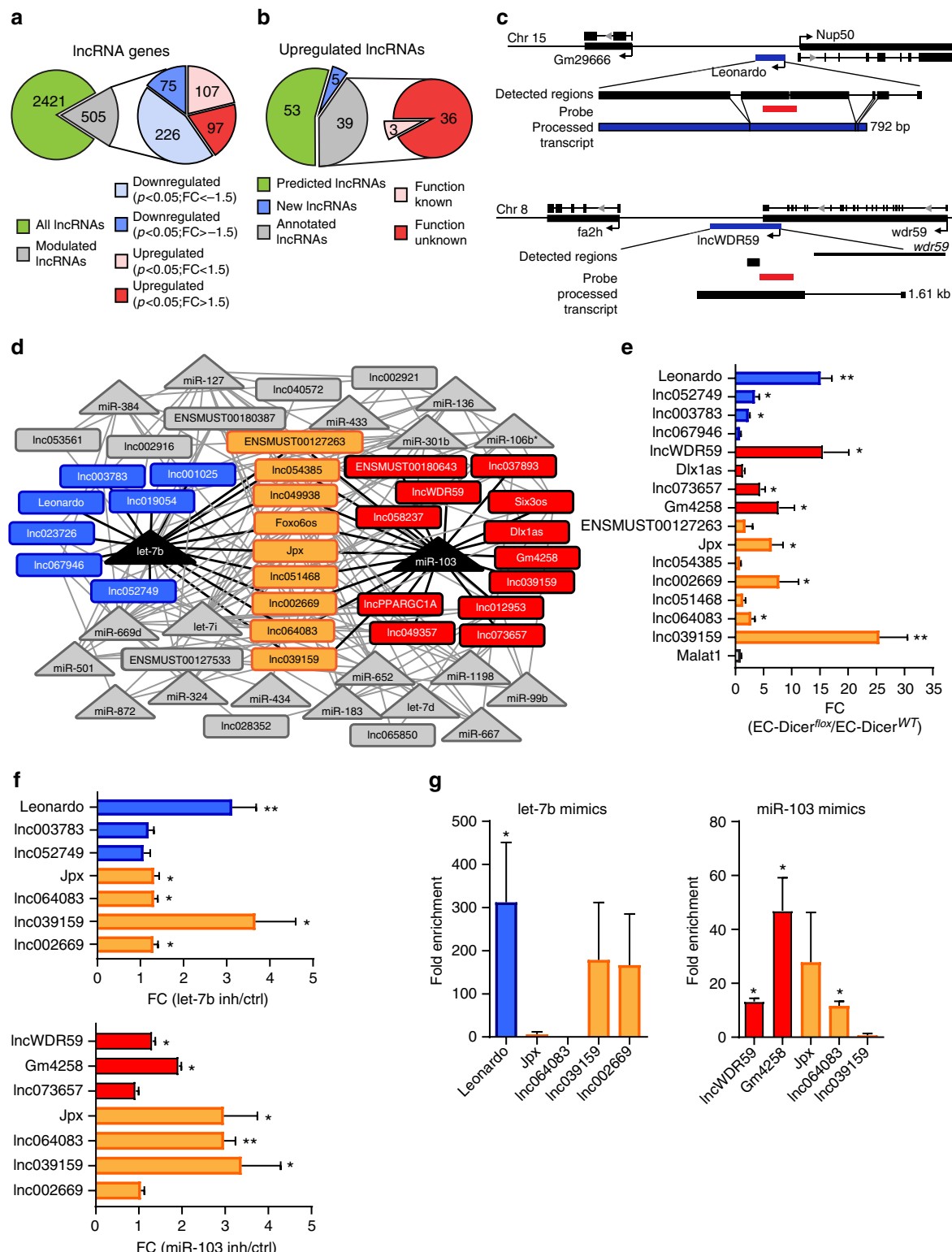

Gm4258, Jpx, and lnc064083 was studied in ECs and murine arteries. In contrast to *lnc064083*, the expression of *lncWDR59*, *Jpx*, and *Gm4258* was higher in MAoECs than in bone marrow-derived macrophages (BMDM) (Fig. 2a). Moreover, lncWDR59 levels were similar between ECs and murine aortic smooth muscle cells (MAoSMCs), whereas miR-103 expression was slightly lower in MAoECs than in ECs and BMDM (Supplementary Figure 4A). Low shear stress, which increases endothelial miR-103 expression[21–25], reduced *lncWDR59* and increased *lnc064083* compared to high shear stress, whereas *Jpx* and *Gm4258* expression was similar in MAoECs exposed to low or high shear stress (Fig. 2b). Treatment of MAoECs with oxLDL, (100 µg ml$^{-1}$) upregulated *miR-103* and decreased *lncWDR59*, *lnc064083*, *Gm4258*, but not *Jpx*, and their expression was restored by inhibition of miR-103 in oxLDL-treated ECs (Supplementary Figure 4A, B and Fig. 2c). Notably, oxLDL treatment did not affect lncWDR59 and miR-103 expression in BMDM and MAoSMCs (Supplementary Figure 4A).

In mice, the expression of *lncWDR59*, *Jpx*, *lnc064083*, and *Gm4258* was substantially higher in athero-protected thoraco-abdominal aortas than in athero-prone aortic arches (Fig. 2d), in opposite to miR-103 expression (Supplementary Figure 4C). Moreover, HFD feeding strongly downregulated *lncWDR59*, *lnc064083* and *Gm4258* and upregulated *miR-103* in the aorta of *Apoe*$^{-/-}$ mice, whereas *Jpx* was not affected (Fig. 2e and Supplementary Figure 4D). In addition, lncRNA expression was studied in laser-microdissected ECs and macrophages from atherosclerotic lesions from *Apoe*$^{-/-}$ mice (Supplementary Figure 4E). *LncWDR59* and *Jpx* expression was higher in ECs than in macrophages, whereas the *lnc064083* and *Gm4258* expression levels were similar in both groups (Fig. 2f). In opposite to the expression of miR-103, in situ hybridization showed that *lncWDR59* was more prominently expressed in ECs from EC-Dicer$^{flox}$ mice than in those from EC-Dicer$^{WT}$ mice (Fig. 2g).

Taken together, these data show that, in contrast to Gm4258, Jpx, and lnc064083, atherogenic conditions such as disturbed flow and hyperlipidemia consistently suppress lncWDR59 in ECs, which is in opposite to the regulation of miR-103.

## miR-103 affects EC proliferation by targeting lncWDR59.

To study the functional role of miR-103 and lncWDR59 interaction, an antisense oligonucleotide (target site blocker (TSB)) was designed to specifically block the binding of miR-103 to its predicted lncWDR59 target site (Fig. 2h). Treatment of MAoECs with TSB increased the expression of *lncWDR59*, but not that of *Jpx*, *Gm4258*, *lnc064083*, or *Klf4* (Fig. 2h). This result confirms the miR-103 target site located at nucleotides 1266–1291 of lncWDR59 transcript (Supplementary Figure 2).

MiR-103 promotes pro-inflammatory chemokine expression by targeting *KLF4*[12]. Although inhibition of miR-103 suppressed *Ccl2* and *Cxcl1* in MAoECs, silencing lncWDR59 or treatment with TSB did not affect the expression of these chemokines (Fig. 3a and Supplementary Figure 4F). Moreover, treatment of HFD-fed *Apoe*$^{-/-}$ mice with TSB that block the interaction between miR-103 and Klf4 did not alter *lncWDR59* expression in arteries (Supplementary Figure 4G). Hence, lncWDR59 does not mediate the pro-inflammatory effects of miR-103 in ECs.

In addition to endothelial inflammation, inhibition of EC proliferation by hyperlipidemia at athero-prone sites promotes atherosclerosis (Supplementary Figure 4H). Accordingly, athero-protective EC-Dicer knockout restored endothelial proliferation (Supplementary Figure 4I). In vitro, treatment of MAoECs with miR-103 inhibitors or TSB increased EC proliferation and reduced cyclin-dependent kinase inhibitor 1a (*Cdkn1a*) expression, whereas lncWDR59 inhibition reduced endothelial replication and upregulated *Cdkn1a* (Fig. 3b). LncWDR59 inhibition did not alter the expression of *Fa2h* and *Wdr59*, indicating that lncWDR59 and not its neighboring genes promotes EC proliferation (Supplementary Figure 4J). Moreover, TSBs promoted cell cycle progression of ECs from the G0 phase to the G1 and S+M phases (Supplementary Figure 4K) and increased proliferation of ECs subjected to low shear stress, but not to high shear stress (Fig. 3c). By contrast, TSB treatment did not affect MAoSMC proliferation (Supplementary Figure 4L). Taken together, these data show that miR-103 inhibits EC proliferation and cell cycle progression by targeting lncWDR59, which may enhance atherosclerosis.

## Notch1 and β-catenin in lncWDR59-induced EC proliferation.

In addition to Klf4, genes related to Wnt and Notch signaling pathways, which can increase cell proliferation[26,27], were most strongly regulated in atherosclerotic arteries from EC-Dicer$^{flox}$ mice[12]. Accordingly, EC-Dicer knockout increased endothelial Notch1 and β-catenin activation in atherosclerotic arteries (Supplementary Figure 5A). Feeding a normal or HFD increased and decreased endothelial Notch1 and β-catenin activation and EC proliferation at predilection sites compared to non-predilection sites (Supplementary Figure 5B-C). In vitro, treatment with the Notch1-inhibitor DAPT or silencing *Ctnnb1* reduced EC proliferation (Fig. 3d and Supplementary

**Fig. 1** LncRNAs upregulated in EC-Dicer$^{flox}$ mice during atherosclerosis and targets of let-7b and miR-103. **a** Pie graph from genome-wide microarray analysis for lncRNA genes differentially expressed in the aortas of EC-Dicer$^{flox}$ mice compared with EC-Dicer$^{WT}$ mice fed 12 weeks of HFD ($n = 2$ mice per group). Not-regulated lncRNA genes are in green, and modulated lncRNAs are in gray. Among these, up- and downregulated lncRNAs with a $P$ value cutoff of 0.05 were divided according to their fold change. **b** Pie graph representing the 97 significantly upregulated lncRNAs, following RNA-seq analysis from MAoECs: 53 unknown lncRNAs (green), 5 lncRNAs for which the sequence was discovered by RNA-seq or RACE-PCR (blue), and 39 lncRNAs with a sequence annotated in NONCODE or ENSEMBL databases (gray). Three out of 36 lncRNAs showed a known function (pink). **c** Gene locus and full transcript sequence for 2 new lncRNAs, upregulated in EC-Dicer$^{flox}$ mice, i.e., *Leonardo* and *lncWDR59*. Detected regions for Leonardo (792 bp) and lncWDR59 (1.61 kb) sequences derive from RNA-seq or RACE-PCR analysis, respectively. Probes from genome-wide microarray are reported in red. **d** Cytoscape interaction network from RNAHybrid binding sites prediction between downregulated miRNAs and upregulated lncRNAs with a determined sequence in EC-Dicer$^{flox}$ mice. LncRNA targets of let-7b or miR-103 are in light blue and red, respectively. LncRNAs with BS for both let-7b and miR-103 are in orange. In gray are the rest of miRNAs target lncRNAs lacking a BS for miR-103 or let-7b. **e, f** The in vivo upregulation of lncRNAs with BS for let-7b and miR-103 was confirmed in EC-Dicer$^{flox}$ mice ($n = 4$–7 mice per group) (**e**) and in MAoECs after let-7b or miR-103 inhibition (**f**) by qPCR ($n = 6$ per group). FC > 2. **g** Fold enrichment of lncRNA transcripts in the RISC complex after transfection of MAoECs with let-7b or miR-103 mimics together with a mutant form of Ago2, following an immunoprecipitation (GW182-IP), measured by qPCR. Results of three independent experiments are shown. GAPDH and B2M were used as housekeeping genes and relative expression analysis. FC fold change of control (ctrl) group. The data are represented as mean ± SEM of the indicated number ($n$) of repeats. *$P < 0.05$, and **$P < 0.01$ by Student's $t$-test

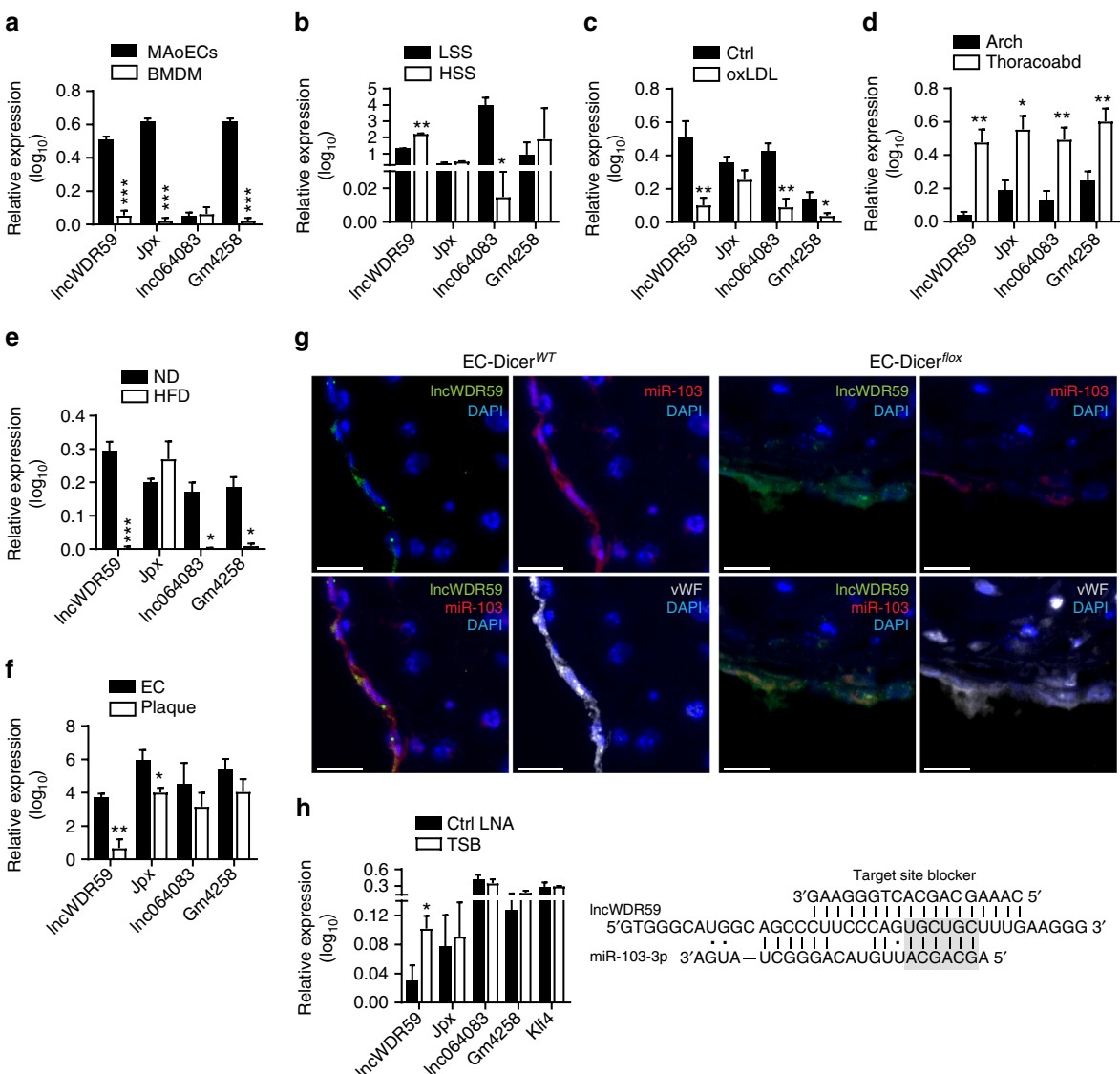

**Fig. 2** Screening of miR-103 target lncRNAs expression and identification of lncWDR59 as target of miR-103. qPCR for miR-103 target lncRNAs expression in different cells or conditions, such as **a** MAoECs and BMDM ($n = 3$–5 per group), **b** MAoECs under high or low shear stress (HSS or LSS) conditions for 48 h ($n = 3$ per group), **c** MAoECs treated with oxidized LDL (oxLDL) for 24 h ($n = 4$ per group), **d** aortic arches (Arch) and thoraco-abdominal (Thoracoabd) aortas from 12-week normal diet-fed $Apoe^{-/-}$ mice ($n = 5$–10 per group), **e** all aorta from $Apoe^{-/-}$ mice fed 12 weeks of normal diet (ND) or high-fat diet (HFD) ($n = 3$–5 per group), and **f** ECs and plaques isolated from $Apoe^{-/-}$ mice fed 4 weeks an HFD using the laser microdissection system ($n = 3$–7 per group). **g** In situ hybridization for miR-103 and lncWDR59 on 12 weeks HFD-fed EC-Dicer$^{WT}$ and EC-Dicer$^{flox}$ mice. vWF and DAPI were used to stain ECs and nuclei, respectively (n = 4 mice per group). **h** The 6-mer seed predicted binding site for miR-103 on lncWDR59 transcript predicted using RNAHybrid and the sequence of the competitive TSB, which specifically inhibits the binding of miR-103 on lncWDR59 transcript ($n = 3$–8 per group). Non-classical Watson and Crick interactions between A and U were represented with a dot. B2m was used as housekeeping gene for relative quantification. EC endothelial cells, MAoECs murine aortic ECs, BMDM bone marrow-derived macrophages, TSB target site blocker. Data are represented as mean ± SEM of the indicated number (n) of repeats. *$P < 0.05$, **$P < 0.01$, and ***$P < 0.001$ by Student's t-test. Scale bar: 25 μm

Figure 5D,E), indicating that Notch1 and Wnt signaling promote athero-protective EC regeneration.

Next, we studied whether Notch1 and β-catenin play a role in lncWDR59-mediated EC proliferation. Inhibition of miR-103 and TSB significantly promoted whereas lncWDR59 knockdown reduced the activation of Notch1 and β-catenin (Fig. 3e, f). TSB-induced EC proliferation was prevented by DAPT, indicating that Notch activation acts downstream of lncWDR59 on endothelial proliferation (Fig. 3g). Surprisingly, silencing *Ctnnb1* enhanced the proliferation of TSB-treated ECs (Fig. 3g).

Moreover, DAPT reduced the proliferation of ECs treated with si*Ctnnb1* and TSB, demonstrating that β-catenin can inhibit EC proliferation by limiting Notch1 activation (Fig. 3h). Accordingly, DAPT decreased β-catenin activation in ECs treated with or without TSBs, whereas silencing *Ctnnb1* reduced Notch1 activation (Notch intracellular domain (NICD)) in control-treated ECs (Fig. 3i, j). Moreover, silencing *Ctnnb1* increased Notch1 activation in TSB-treated ECs (Fig. 3i, j). These data show that the interaction between miR-103 and lncWDR59 inhibits EC proliferation due to reduced lncWDR59-mediated

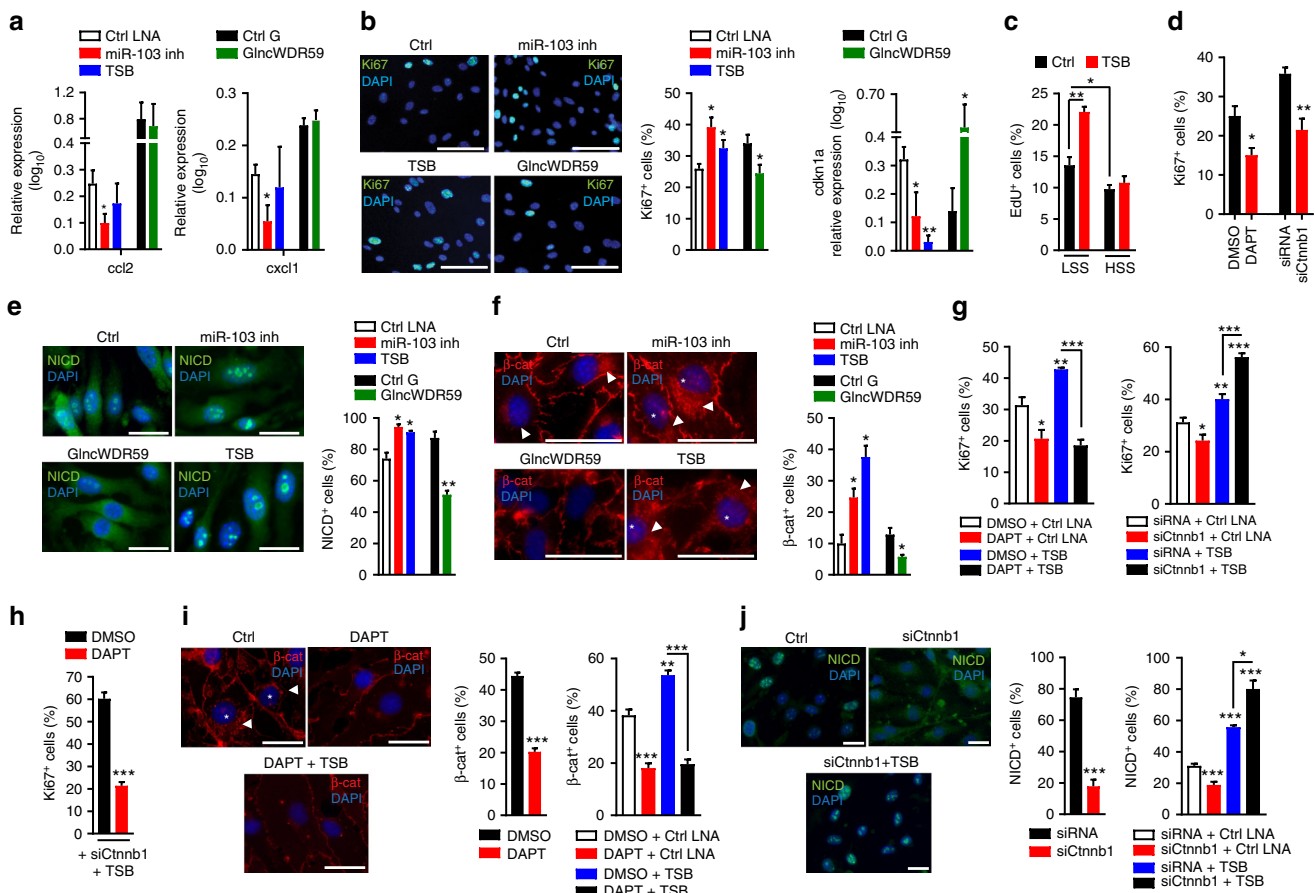

**Fig. 3** miR-103 target lncWDR59 regulates EC proliferation through Wnt and Notch1 signaling pathways. MAoECs were transfected for 24 h with miR-103 inhibitors (miR-103 inh), lncWDR59 Gapmers (GlncWDR59), or TSB (**a**) to check Ccl2 and Cxcl1 relative expression (n = 3–6 per group), or (**b**) to stain with an anti-Ki67 antibody to analyze their proliferation and cdkn1a expression by qPCR (n = 4–8 per group). DAPI was used for nuclei staining. EC proliferation was calculated as number of Ki67+ cells on total number of cells and expressed in percentage (n = 4 per group). **c** MAoECs were seeded in ibidi flow chamber slides and incubated with EdU and TSB or control LNAs. ECs were subjected for 48 h to low (LSS; 5.51 dyn cm$^{-2}$) or high (HSS; 10 dyn cm$^{-2}$) shear stresses and stained for EdU-DNA incorporation (n = 3–4 per group). **d** MAoECs were treated with DAPT or siRNAs against β-catenin (siCtnnb1) for 24 h and stained with anti-Ki67 antibody to analyze their proliferation as described before. DMSO and siRNA were used as controls of DAPT and siCtnnb1, respectively (n = 4–8 per group). **e, f** MAoECs were transfected with miR-103 inhibitors, TSBs, or GlncWDR59 as described before and stained with an anti-activated Notch1 (NICD) (**e**) or β-catenin antibody (**f**). Data were represented as number of nuclear NICD+, and nuclear (star) and perinuclear (arrowheads) β-cat+ localization. Inhibition of β-catenin activation makes β-catenin visible only as membrane adaptor protein for cell-to-cell intercellular adhesions. Data were normalized on total number of cells and expressed in percentage (n = 4–8 per group). **g–j** MAoECs were treated for 24 h with DAPT or siCtnnb1, alone or in combination with TSB to analyze (**g**) EC proliferation by Ki67 staining. DMSO or siRNAs in combination with LNA-controls were used as controls. Data were normalized on total number of cells and expressed in percentage (n = 4 per group). **h** Analysis of Ki67+ TSB +siCtnnb-transfected MAoECs stimulated with DAPT or DMSO for 24 h (n = 4 per group). Analysis of β-catenin (**i**) or NICD (**j**) nuclear localization. The graphs correspond to the analysis of nuclear (stars) and perinuclear (arrowheads) β-cat+ and nuclear NICD+ localization. Data were normalized on total number of cells and expressed in percentage (n = 4–8 per group). Ccl2 C-C motif chemokine ligand 2, Cxcl1 C-X-C motif chemokine ligand 1, cdkn1a cyclin-dependent kinase inhibitor 1a, TSB target site blockers, DMSO dimethyl sulfoxide, DAPT γ-secretase inhibitor. Data are represented as mean ± SEM of the indicated number (n) of repeats. *P < 0.05, **P < 0.01, and ***P < 0.001 by one-way ANOVA and t-test. Scale bar: 25 μm

Notch1 and β-catenin activation. Moreover, blocking miR-103-mediated repression of lncWDR59 induced a negative feedback loop in which β-catenin limited Notch1 activation and EC proliferation.

**lncWDR59 blocks Numb to promote Notch1 activation.** To study the mechanisms by which lncWDR59 regulates Notch1 activation, the subcellular localization of lncWDR59 was analyzed. Treatment of MAoECs with miR-103 inhibitors increased the cytosolic levels of lncWDR59 (Fig. 4a), indicating that lncWDR59 may bind to Notch1 or to cytosolic proteins related to the Notch1 signaling pathway. Therefore, we determined the

interaction propensity between lncWDR59 and NICD, deltex4, Numb, anterior pharynx-defective 1 (aph-1), presenilin-1 and -2 (psen-1 and -2), and nicastrin using the catRAPID prediction software. LncWDR59 had no binding sites predictable in the amino acid sequence of NICD. Accordingly, lncWDR59 was not enriched in NICD-immunoprecitiated (IP) samples from MAoECs treated with control LNAs or miR-103 inhibitors (Fig. 4b and Supplementary Figure 6A), indicating that lncWDR59 does not bind to NICD. Among the Notch1 pathway-related proteins, Numb contained sites with the highest propensity to interact with lncWDR59, (Fig. 4c and Supplementary Figure 1F). To study whether lncWDR59 binds to Numb, we performed Numb-IP, which revealed that lncWDR59 was not

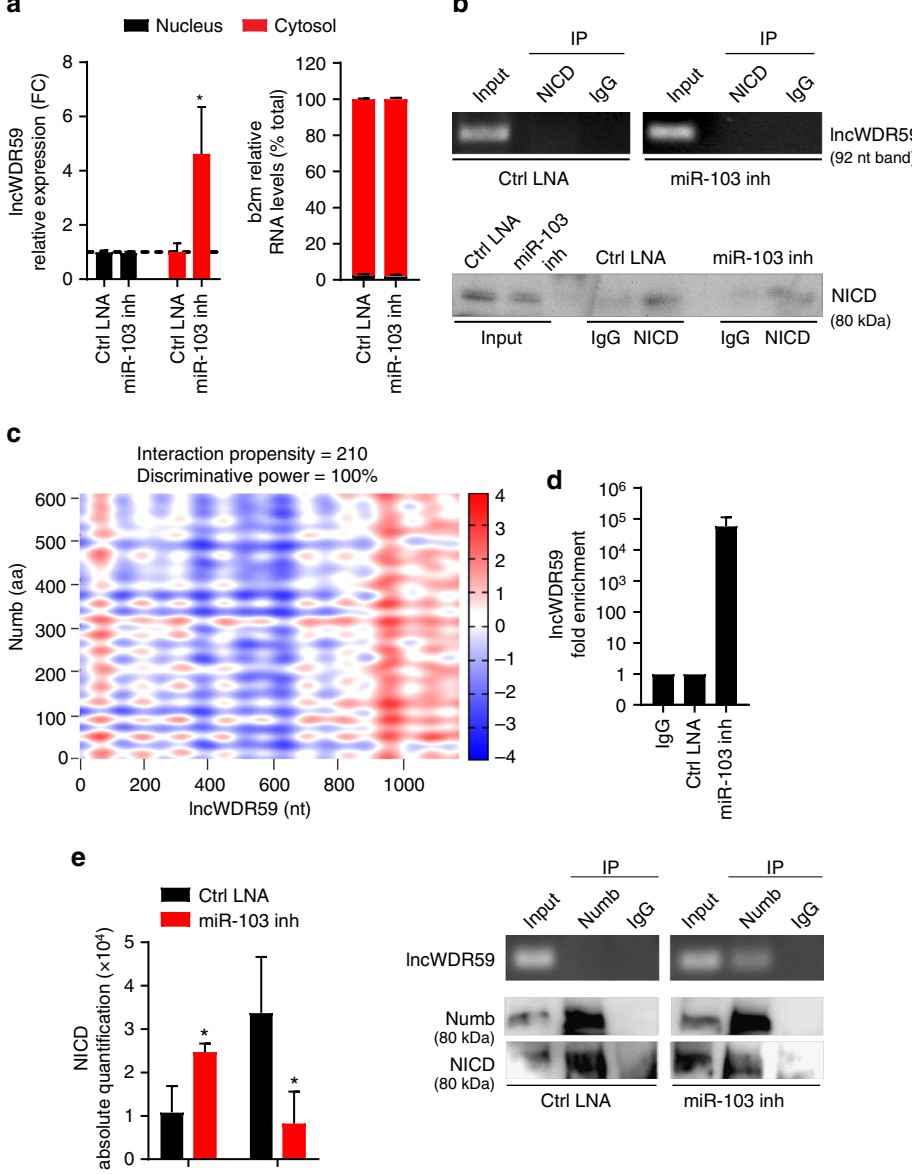

**Fig. 4** LncWDR59 interaction with Notch1-inhibitor Numb. **a** Nuclear and cytoplasmic lncWDR59 levels in MAoECs treated with miR-103 or control-LNA inhibitors for 24 h. B2m was used as quality marker of nucleus and cytoplasmic RNA isolation ($n = 3$ per group), expressed as percentage of total b2m RNA expression. Gapdh was used for relative quantification. **b** LncWDR59 expression in MAoECs before (input) and after NICD-immunoprecipitation (IP) analyzed by qPCR and loaded on 2% agarose gel (lncWDR59 amplicon of 92 bp). Efficiency of NICD-IP was tested by western blot ($n = 3$ per group). IgGs were used as IP-negative controls. **c** Heat-maps representing the interaction propensity analysis between lncWDR59 and Numb. The $y$- and $x$-axes represent the index of the protein and lncWDR59 sequences, respectively. The colors indicate the interaction score of individual nucleotide and amino acid pairs (rank ± 4). The interaction strength with respect to a training set is represented by the interaction score and the discriminative power values. **d**, **e** LncWDR59 expression in MAoECs treated with miR-103 or control-LNA inhibitors for 24 h analyzed by qPCR, following Numb-IP. Amplification products were loaded on 2% agarose gel (lncWDR59 amplicon of 92 bp), together with PCR products from lncWDR59 expression analyzed before IP (**d**). Numb-IP efficiency was evaluated by western blot (**d**) and NICD binding to Numb quantified ($n = 3$ per group) (**e**). IgGs were used as IP-negative controls. FC fold change of control-LNA inhibitors of the corresponded subcellular fraction, IP immunoprecipitate, NICD Notch intracellular domain, aa amino acid, nt nucleotides. *$P < 0.05$ by Student's $t$-test

enriched (Supplementary Figure 6B). However, NICD was high in the Numb-IP fraction, whereas it was low in the input fraction (Fig. 4d, e and Supplementary Figure 6C, D). To test the existence of a possible competitive binding between lncWDR59 and NICD to Numb, Numb-IP was performed in MAoECs after miR-103 inhibition. LncWDR59 was highly enriched in the Numb-IP compared to the input fraction. In addition, miR-103 inhibition

reduced NICD levels in the Numb-IP fraction and increased those in the input (Fig. 4d, e and Supplementary Figure 6B–D). Interestingly, one of the four predicted Numb-interaction sites on lncWDR59 transcript comprises miR-103 BS, and involves the phosphotyrosine-binding domain of Numb, through which Numb interacts and induces Notch1 degradation[28]. Hence, these data suggest that binding of miR-103 on lncWDR59 prevents

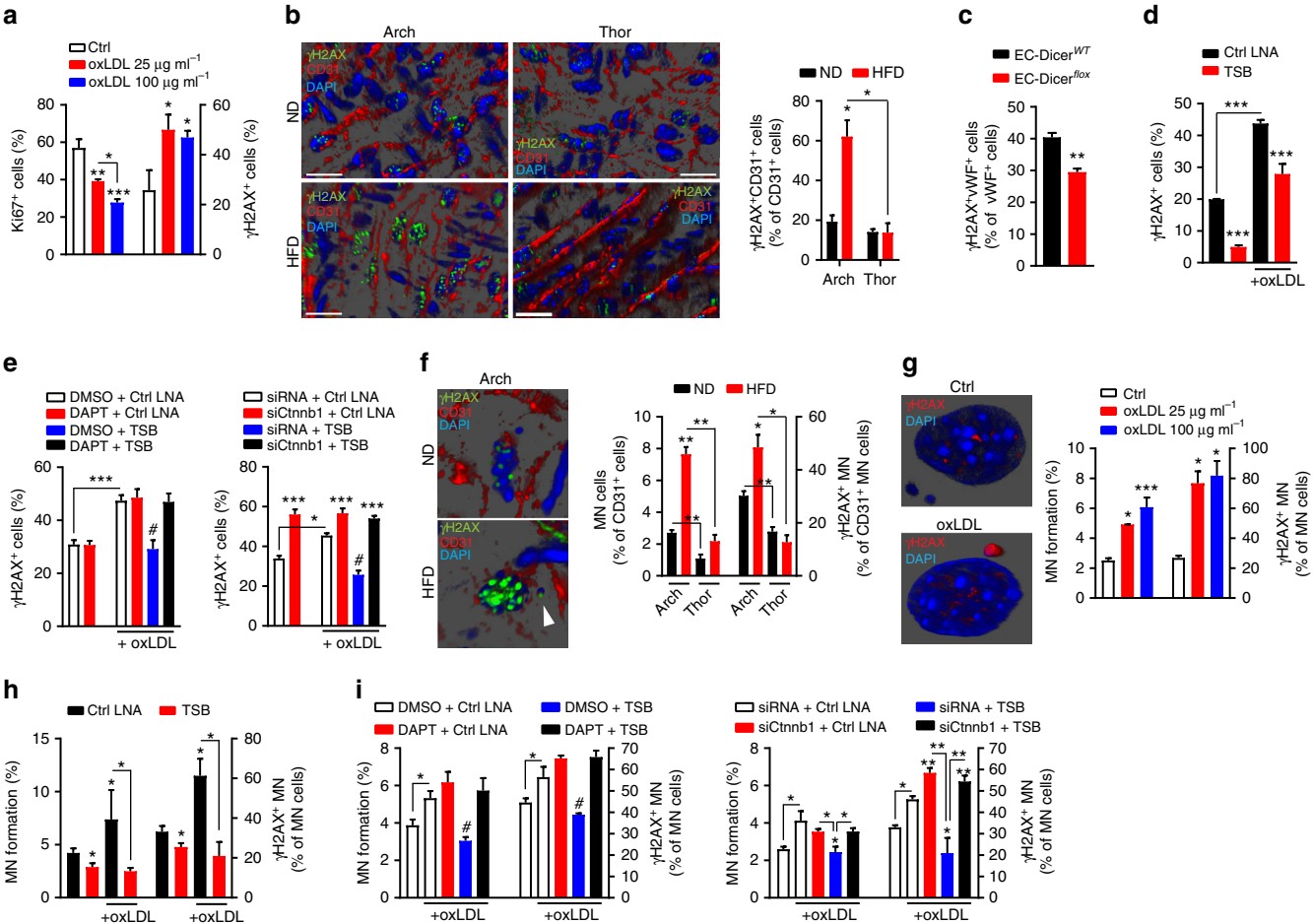

**Fig. 5** Role of lncWDR59 on hyperlipidemia-mediated DNA damage and MN formation in ECs. **a** Immunofluorescence analysis of Ki67 or phosphorylated gamma H2AX histone (γH2AX) in MAoECs treated for 24 h with 25 or 100 μg ml$^{-1}$ oxLDL. Data are represented as percentage of total number of cells ($n$ = 4 per group). **b** The en face 3D reconstructed arch and thoracic aortae from $Apoe^{-/-}$ mice fed 12 weeks of ND or HFD and stained for γH2AX and CD31. The graph represents the number of CD31$^+$ cells with a positive γH2AX staining in the nucleus, normalized on total number of CD31$^+$ cells and expressed in percentage ($n$ = 4 mice per group). **c** Analysis of nuclear γH2AX$^+$ vWF$^+$ in the root of EC-Dicer$^{flox}$ mice compared to EC-Dicer$^{WT}$ mice, fed 12 weeks of HFD, and expressed as percentage of total vWF$^+$ cells ($n$ = 3 mice per group). **d** Analysis of nuclear γH2AX$^+$ MAoECs treated with TSBs for 24 h, and stimulated with or without 25 μg ml$^{-1}$ oxLDL. Data are normalized on total number of ECs and expressed in percentage ($n$ = 4–6 per group). **e** MAoECs were treated for 24 h with DAPT or siCtnnb1 to analyze the nuclear γH2AX$^+$ staining. The same treatments were performed in 25 μg ml$^{-1}$ oxLDL-treated MAoECs, alone or in combination with TSB, to analyze the nuclear γH2AX$^+$ staining. Data are normalized on total number of cells and expressed in percentage ($n$ = 3–5 per group). **f** Analysis of micronucleated CD31$^+$ cells (MN cells) and CD31$^+$ cells with γH2AX$^+$ micronuclei (γH2AX$^+$ MN) on en face 3D reconstructed arch and thoracic aortae from $Apoe^{-/-}$ mice fed 12 weeks of ND or HFD ($n$ = 3 mice per group). **g** Analysis of MN formation and γH2AX$^+$ MN from micronucleated MAoECs (γH2AX$^+$ MN) treated for 24 h with 25 or 100 μg ml$^{-1}$ oxLDL ($n$ = 3 per group). Micronucleated MAoECs treated or not with 25 μg ml$^{-1}$ oxLDL, captured using confocal microscope. Transversal section shows γH2AX staining around the heterochromatin of principal nucleus and inside of the MN. **h** Analysis of MN formation and γH2AX$^+$ MN in micronulceated MAoECs treated with TSBs for 24 h, and stimulated with or without 25 μg ml$^{-1}$ oxLDL. Data are normalized on total number of ECs and expressed in percentage ($n$ = 4–10 per group). **i** Analysis of MN formation and γH2AX$^+$ MN MAoECs treated for 24 h with DAPT or siCtnnb1, in combination with 25 μg ml$^{-1}$ oxLDL ($n$ = 3–10 per group). DMSO dimethyl sulfoxide, DAPT γ-secretase inhibitor. Data are represented as mean ± SEM of the indicated number ($n$) of repeats. *$P$ < 0.05, **$P$ < 0.01, and ***$P$ < 0.001 by Student's $t$-test. #$P$ < 0.05 versus all other groups by one-way ANOVA and two-way ANOVA. Scale bar: 25 μm

lncWDR59 and Numb interaction, allowing Numb interaction to NICD.

**lncWDR59 reduces oxLDL-induced DNA damage**. EC proliferation regenerates low shear stress-induced damage of ECs[6]. However, additional injuries, for example oxLDL-induced DNA damage[29], may exhaust the EC repair capacity and enhance atherosclerosis[6]. In line, treatment of MAoECs for 24 h with oxLDL dose-dependently increased DNA damage and reduced EC proliferation determined by immunostaining of

phosphorylated histone variant H2AX (γH2AX foci)[30] and Ki67, respectively (Fig. 5a). Moreover, we performed combined γH2AX and CD31 immunostaining of en face prepared aortas from $Apoe^{-/-}$ mice by image stack and three-dimensional (3D) reconstruction microscopy to determine the effect of HFD on endothelial DNA damage. Whereas endothelial DNA damage was not different between predilection and non-predilection sites in ND-fed mice, a 12-week HFD increased γH2AX in ECs at predilection sites, but not at non-predilection sites of atherosclerosis (Fig. 5b). This result indicates that low shear stress increases the

susceptibility of ECs to oxLDL-induced DNA damage. Hence, hyperlipidemia may promote atherosclerosis by limiting EC proliferation and by increasing endothelial DNA damage. Notably, endothelial Dicer knockout reduced endothelial γH2AX in hyperlipidemic $Apoe^{-/-}$ mice (Fig. 5c), indicating a role of miRNAs in hyperlipidemia-induced endothelial DNA damage. To study whether miR-103 affects endothelial oxLDL-mediated DNA damage by targeting lncWDR59, γH2AX was analyzed in MAoECs treated with TSBs and oxLDL. TSBs reduced γH2AX in MAoECs treated with or without oxLDL (Fig. 5d). Blocking Notch1 by DAPT or silencing $Ctnnb1$ abolished the TSB-mediated reduction of γH2AX in oxLDL-treated

ECs (Fig. 5e). However, in the absence of TSB silencing, $Ctnnb1$ but not DAPT increased γH2AX in ECs treated with or without oxLDL (Fig. 5e and Supplementary Figure 7A). Taken together, these data indicate that the protective effect of lncWDR59 on endothelial DNA damage is mediated by Notch1 and β-catenin activation.

**lncWDR59 reduces oxLDL-mediated micronuclei formation.** During proliferation the chance of mitotic errors is high and can lead to the generation of extranuclear chromatin bodies, called micronuclei (MN), where double-strand breaks accumulate[31,32].

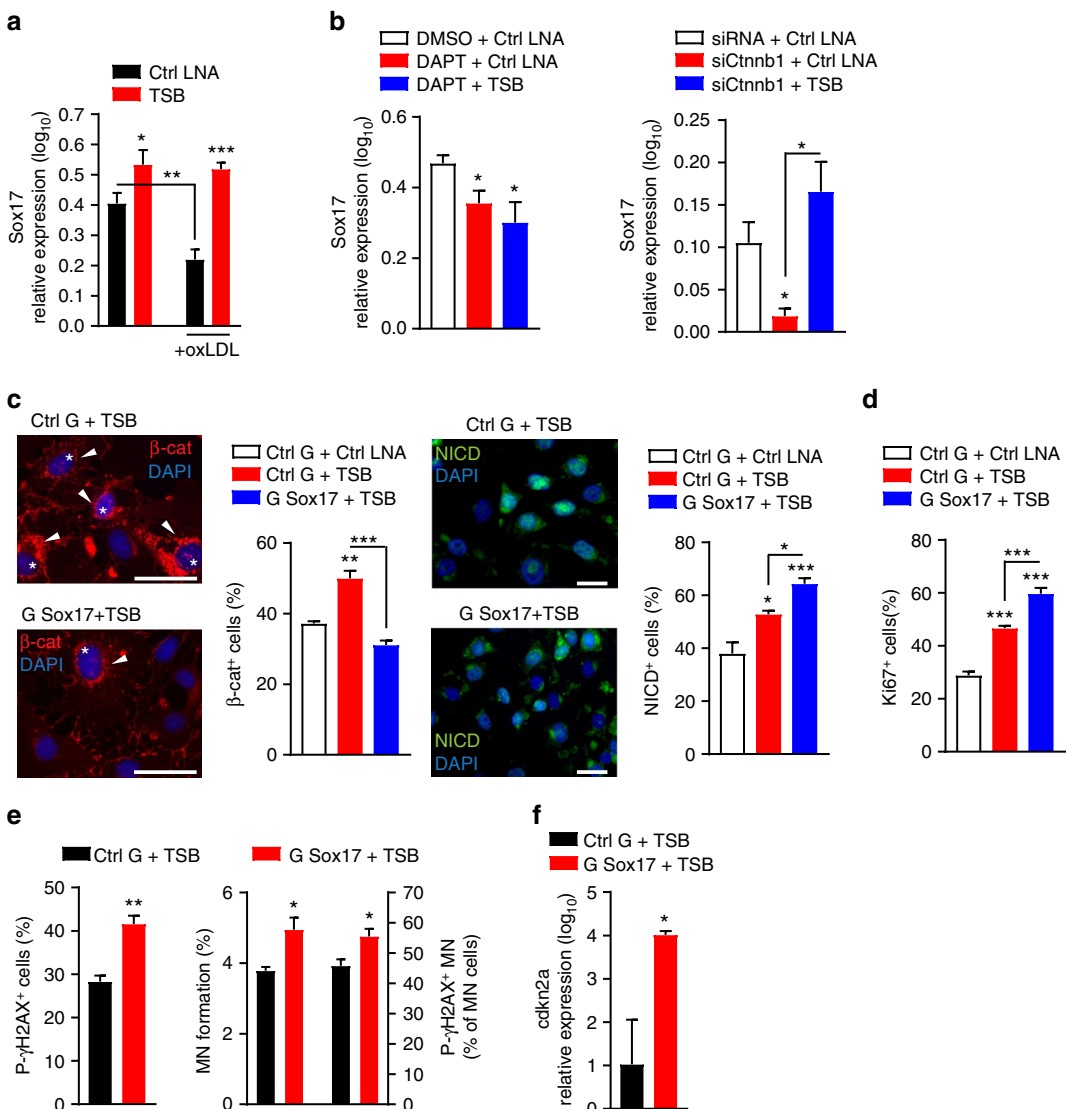

**Fig. 6** Role of Sox17 on β-catenin activation, MN formation, and micronuclear DNA damage. **a** qPCR analysis of Sox17 expression in MAoECs treated with TSB, alone or in combination with 100 μg ml$^{-1}$ oxLDL for 24 h ($n = 4$ per group). **b** qPCR analysis of Sox17 expression in MAoECs treated for 24 h with DAPT or siCtnnb1, alone or in combination with TSB ($n = 4$ per group). B2m was used for relative quantification. **c, d** Analysis of nuclear (stars) and perinuclear (arrowheads) β-catenin, activated Notch intracellular domain (NICD) (**c**), or nuclear Ki67 staining (**d**) in MAoECs treated for 24 h with Sox17 gapmers, alone or in combination with TSB. Data are normalized on total number of cells and expressed in percentage ($n = 4$ per group). **e, f** Analysis of γH2AX+ cells, MN formation, γH2AX$^+$ MN from micronucleated cells (γH2AX$^+$ MN) (**e**), and cdkn2a expression (**f**) in MAoECs treated for 24 h with Sox17 gapmers, alone or in combination with TSB ($n = 4$–6 per group). B2m was used for relative quantification. Data regarding MN formation are normalized on total number of cells and expressed in percentage. Data regarding γH2AX$^+$ MN are normalized on total number of micronucleated cells, following normalization on total number of cells, and expressed in percentage. TSB target site blocker, DMSO dimethyl sulfoxide, DAPT γ-secretase inhibitor, G Sox17 Sox17 gapmers, cdkn2a cyclin-dependent kinase inhibitor 2a. *$P < 0.05$, **$P < 0.01$, and ***$P < 0.001$ by Student's $t$-test. By two-way ANOVA and one-way ANOVA. Scale bar: 25 μm

In line, the number of MN was high in proliferating ECs (Supplementary Figure 7B).

Whereas damaged DNA is efficiently repaired in the nucleus, persistent DNA damage in the MN may further impair transcriptional activity in MN and compromise cell function[32]. Because in vitro oxidative stress can induce MN in ECs, we analyzed the effect of hyperlipidemia on MN formation in ECs in arterial en face prepared aortas. In contrast to the thoracic aorta, where MN-containing ECs were less than in the aortic arch under ND, HFD feeding increased MN formation in aortic arch ECs (Fig. 5f). Moreover, in ND-fed mice micronuclear γH2AX in aortic arch ECs was more frequent than in thoracic ECs (Fig. 5f). HFD feeding increased the micronuclear γH2AX in aortic arch but not thoracic ECs (Fig. 5f). In vitro, oxLDL increased MN formation and micronuclear γH2AX (Fig. 5g). These data indicate that oxLDL induces MN formation and micronuclear DNA damage in ECs at predilection sites of atherosclerosis.

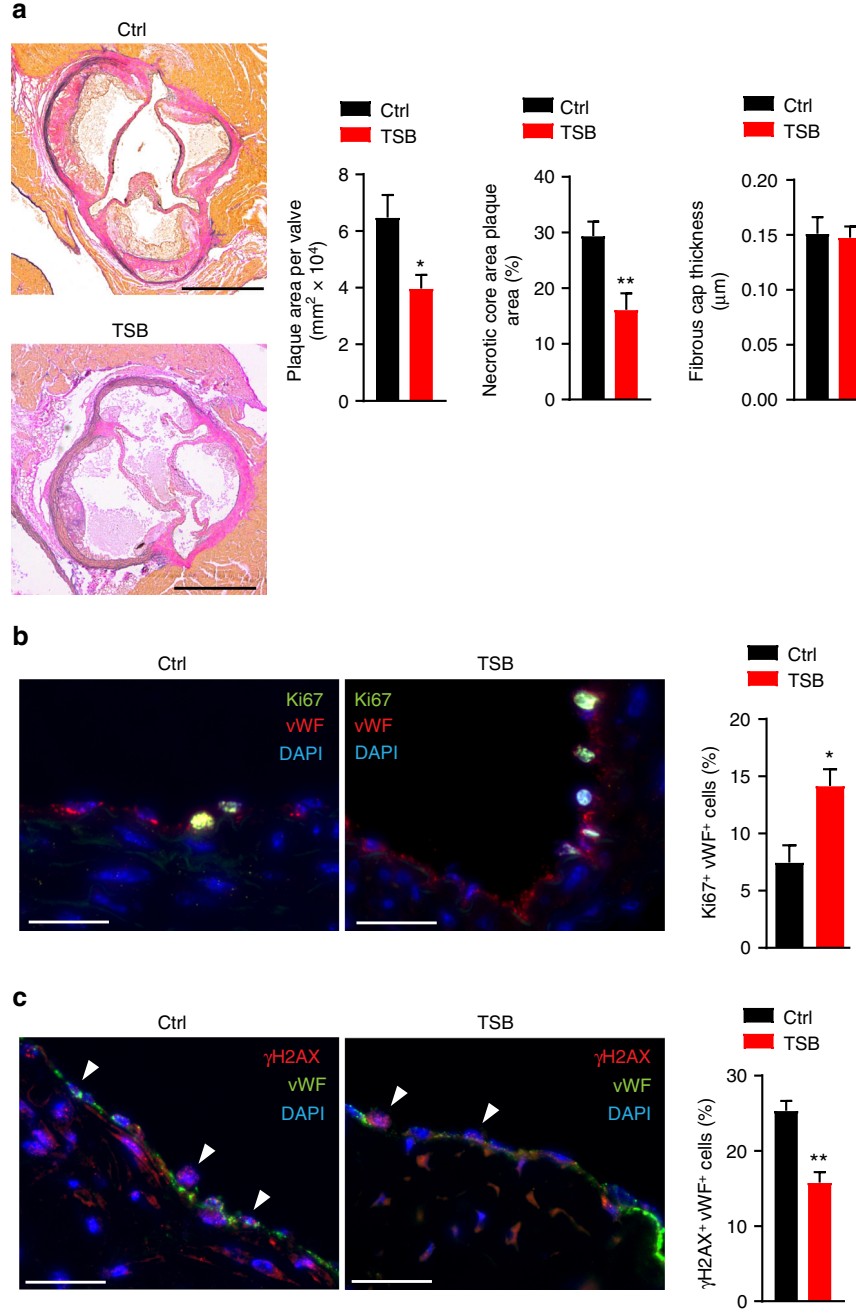

**Fig. 7** MiR-103/lncWDR59 conserved function in vivo during atherosclerosis. $Apoe^{-/-}$ mice were fed 12 weeks of HFD and were injected four times (once per week) with TSBs or control-LNA oligonucleotides (0.5 mg per 20 g). Paxgene-fixed/paraffin-embedded roots were sectioned and used for (**a**) elastic van Gieson (EVG) staining, (**b**) Ki67, and (**c**) γH2AX immunofluorescence staining. **a** EVG-stained aortic roots were used to quantify the plaque area per valve, the necrotic core area (expressed as percentage of total plaque area), and the fibrous cap thickness ($n = 5$ mice per group, 3–4 slides per mouse). Scale bar: 200 μm. **b,c** The number of Ki67+/ and γH2AX+/vWF+ cells was divided to the total number of vWF+ cells and expressed as percentage ($n = 5$ mice per group, 3 slides per group). Ctrl control-LNA, TSB target site blockers. *$P < 0.05$ and **$P < 0.01$ by Student's $t$-test. Scale bar: 25 μm

Next, we studied the role of miR-103-mediated suppression of lncWDR59 on oxLDL-induced MN formation in ECs. TSBs abolished oxLDL-induced MN formation and micronuclear γH2AX (Fig. 5h), an effect prevented by blocking Notch1 or *Ctnnb1* silencing (Fig. 5i). Moreover, in the absence of TSB, silencing *Ctnnb1* but not DAPT increased MN formation and micronuclear γH2AX in EC treated with or without oxLDL (Fig. 5i and Supplementary Figure 7C, D). Hence, these results suggest that suppression of lncWDR59 by miR-103 in ECs increases MN formation and micronuclear DNA damage in response to oxLDL by inhibiting Notch1 and β-catenin. Moreover, hyperlipidemia may promote atherosusceptibility of predilection sites by DNA damage accumulation in the endothelial MN.

Analysis of cyclin-dependent kinase inhibitor 2a (*cdkn2a*, $p16^{INK4A}$) expression showed that treatment of MAoECs with TSB reduced *Cdkn2a*, whereas DAPT treatment or *Ctnnb1* silencing increased and reduced *Cdkn2a*, respectively (Supplementary Figure 7E). Moreover, DAPT treatment or *Ctnnb1* silencing abolished and enhanced TSB-mediated effect on *Cdkn2a* expression (Supplementary Figure 7E), suggesting that lncWDR59 might protect ECs from DNA damage-associated premature senescence[8,33].

**lncWDR59 activates β-catenin through Notch1-mediated Sox17 expression**. To determine the mechanism by which lncWDR59 promotes Notch1-mediated β-catenin activation, we studied genes related to the Wnt and Notch1 signaling pathways differentially regulated in HFD-fed EC-Dicer*flox* mice[12], such as lymphoid enhancer binding factor 1 (Lef1), dickkopf WNT signaling pathway inhibitor 2 (Dkk2), sex determining region Y-box (Sox) 17 (Sox17), deltex E3 ubiquitin ligase 4 (Dtx4), Sox4, and Numb (Supplementary Figure 7F). Inhibition of miR-103 increased *Lef1*, *Dkk2*, *Sox17*, and *Dtx4* in MAoECs, but did not affect *Sox4* and *Numb* (Supplementary Figure 7F). LncWDR59 inhibition reduced *Sox17*, *Dtx4*, and *Sox4* and increased *Dkk2*, whereas *Lef1* and *Numb* expression was unchanged (Supplementary Figure 7F). Notably, TSBs increased *Sox17* expression in MAoECs treated with or without oxLDL, and prevented the oxLDL-induced downregulation of *Sox17* (Fig. 6a). Although *Sox17* expression was decreased by both DAPT and *Ctnnb1* silencing, TSBs prevented only the effect of *Ctnnb1* silencing on *Sox17* expression (Fig. 6b). Silencing *Sox17* abolished the activation of β-catenin induced by TSB, whereas TSB-mediated Notch1 activation was slightly increased after silencing *Sox17* (Fig. 6c). Inhibition of *Sox17* enhanced EC proliferation, *cdkn2a* expression, γH2AX, MN formation, and micronuclear γH2AX in TSB-treated MAoECs (Fig. 6d-f). Taken together, these data indicate that lncWDR59 promotes β-catenin activity by upregulating Notch1-mediated Sox17 expression. Moreover, like β-catenin, Sox17 limits EC proliferation and reduces micronucleic DNA damage.

**Targeting of lncWDR59 by miR-103 promotes atherosclerosis**. To assess the role of the interaction between miR-103 and lncWDR59 in vivo, *Apoe*$^{-/-}$ mice were treated with lncWDR59-TSBs or control LNA-modified oligonucleotides (0.5 mg per 20 g per injection) during the last 4 weeks of a 12-week HFD feeding program. Compared with mice treated with control LNAs, treatment with lncWDR59-TSBs reduced atherosclerosis, the necrotic core area, and endothelial DNA damage, and increased EC proliferation (Fig. 7a–c), whereas the fibrous cap thickness and the Cxcl1-expressing ECs (Supplementary Figure 7H) were not affected (Fig. 7a). Taken together, these data indicate that

miR-103 enhances atherosclerosis and impairs EC regeneration partly by suppressing lncWDR59.

**Conserved function of human lncWDR59 in ECs**. In humans, a homolog of lncWDR59 (hsa-lncWDR59) exists expressed from a conserved genomic location between the *FA2H* and *WDR59* genes on human chromosome 16 (Fig. 8a). MiR-103 mimics enriched hsa-lncWDR59 in the GW182-IP fraction from human aortic ECs (HAoECs) compared to control mimics (Fig. 8b). Transfection of HAoECs with hsa-lncWDR59 inhibitors downregulated hsa-lncWDR59 expression (Supplementary Figure 7G), reduced EC proliferation, and MN formation (Fig. 8c).

In human atherosclerotic lesions, in situ hybridization indicated that endothelial expression of miR-103 and lncWDR59 was high and low, respectively, in human atherosclerotic lesions compared with control vessels (Fig. 8d). Analysis of hsa-lncWDR59 expression in human atherosclerotic lesions indicated that hsa-lncWDR59 and SOX17 expression levels were negatively correlated with necrotic core area and positively with endothelial proliferation, whereas the levels of *CDKN2A* were correlated with increased necrotic core area (Fig. 8e and Supplementary Figure 7I). Moreover, the levels of *hsa-lncWDR59* correlated with increased *SOX17* levels (Fig. 8e). Taken together, these data indicate that hsa-lncWDR59 shows a conserved function in human ECs and might also play an athero-protective role in humans.

## Discussion

Chronic, low-grade endothelial injury and regeneration is a characteristic feature of endothelial maladaptation to disturbed flow at arterial bifurcations. We found that miR-103 inhibits endothelial proliferation by targeting lncWDR59, which resulted in reduced activation of Notch1 signaling. Moreover, suppression of lncWDR59 by miR-103 increased the susceptibility of proliferating ECs to oxLDL-mediated DNA damage and mitotic errors by reducing Notch1-mediated β-catenin activation. Hence, miR-103-mediated suppression of lncWDR59 promotes endothelial maladaptation by impairing EC regeneration and increasing mitotic aberrations during hyperlipidemia (Supplementary Figure 8).

Endothelial Dicer promotes arterial susceptibility to atherosclerosis by generating miRNAs[9,10,12]. In addition to mRNAs, miRNAs can interact with other RNA species, like lncRNA transcripts. This interaction can inhibit miRNA function, because lncRNAs can act as molecular decoys or sponges of miRNAs[34,35] or mediate the silencing of lncRNAs[36,37]. Our data show that Dicer suppressed the expression of multiple lncRNAs in ECs, indicating that targeting of lncRNAs by miRNAs plays a role in EC maladaptation. The miRNAs miR-103 and let-7b silenced lncRNAs, such as the novel intergenic lncRNAs lncWDR59 and Leonardo, respectively, which contain mainly non-canonical miRNA binding sites. We confirmed that miR-103 targets lncWDR59, which was downregulated at predilection sites and by hyperlipidemia, via a 6-mer binding site in ECs. Whereas miR-103 promotes endothelial inflammation by silencing KLF4[12], the interaction between miR-103 and lncWDR59 in ECs inhibited cell proliferation and, in response to oxLDL, increased micronucleic DNA damage. Hence, miR-103 might promote an EC maladaptive phenotype by the synergistic silencing of Klf4 and lncWDR59. In line, our data indicate that lncWDR59 limits the increased susceptibility of proliferating ECs to oxLDL-induced DNA damage and micronuclei formation. Moreover, miR-103-induced silencing of lncWDR59 may mediate the effect of endothelial Dicer on disturbed flow-induced EC proliferation and

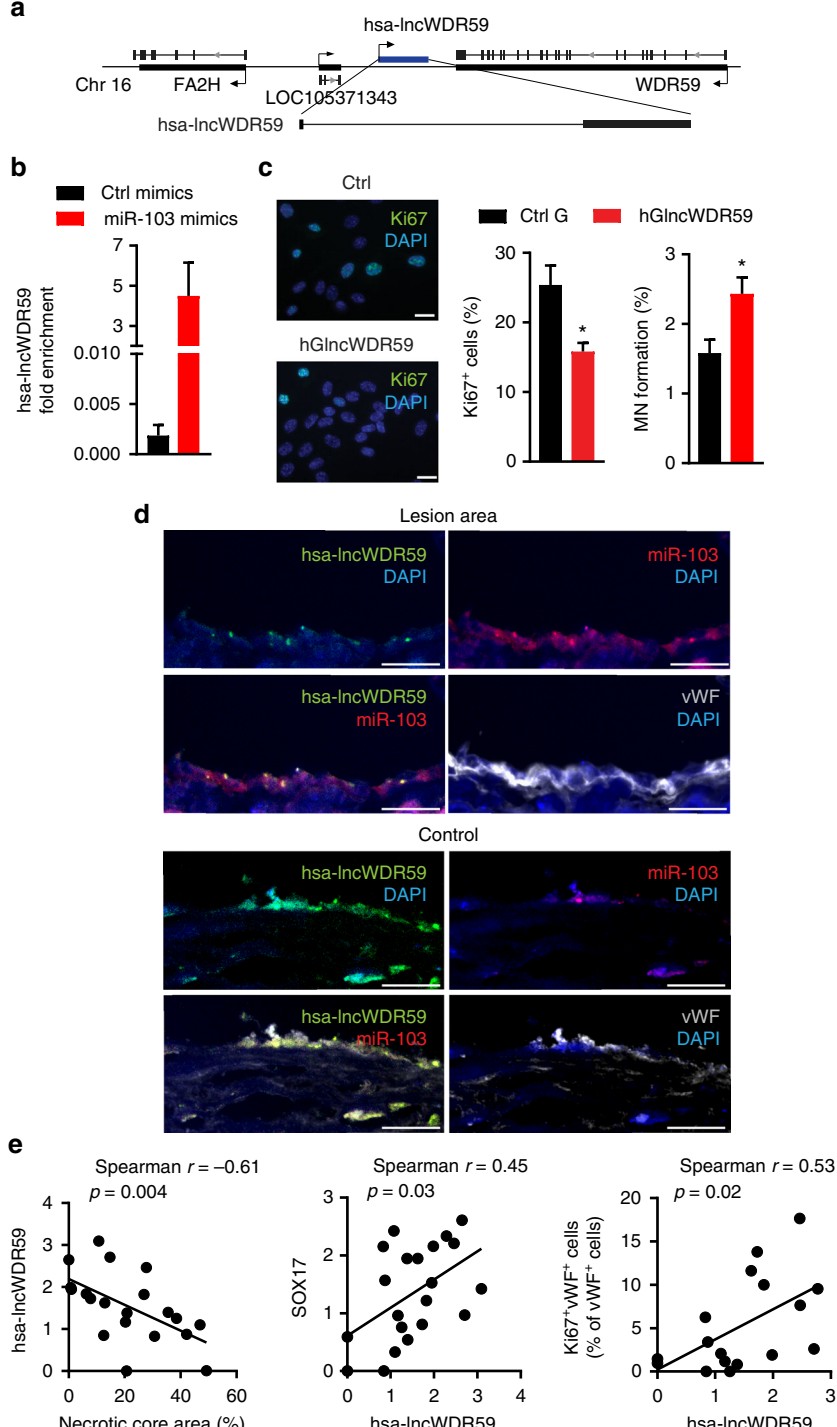

**Fig. 8** Human lncWDR59 conserved role in vitro and in human plaques. **a** Gene locus and full transcript of human lncWDR59 (hsa-lncWDR59) sequence on chromosome 16 between FA2H and WDR59 genes. **b** Enrichment of hsa-lncWDR59 transcripts in the RISC complex after transfection of human aortic ECs (HAoECs) with miR-103 or control mimics together with a mutant form of Ago2, following an immunoprecipitation of GW182 protein (GW182-IP). Results of three experiments are shown. GAPDH was used as housekeeping gene and for relative expression analysis. **c** Analysis of Ki67 immunostaining and MN formation in HAoECs transfected for 48 h with hsa-lncWDR59 gapmers (hGlncWDR59). Data are normalized and expressed as percentage of the total number of cells ($n = 4$ per group). **d** In situ hybridization for miR-103 and lncWDR59 on human plaques. Areas lacking of lesions were used as control areas. vWF and DAPI were used to stain ECs and nuclei, respectively ($n = 4$ per group). **e** Correlation of the relative expression levels of hsa-lncWDR59 in human carotid lesions with necrotic core area, SOX17 expression, and Ki67 endothelial staining (Ki67[+]vWF[+]), ($n = 17$–20 per group). *$P < 0.05$ by Student's $t$-test and a compute nonparametric Spearman's rho correlation analysis. Scale bar: 25 μm

hyperlipidemia-induced EC injury and, together with the targeting of Klf4, promote atherosclerosis.

Previous studies indicate that endothelial Notch1 signaling triggers endothelial differentiation[38,39], promotes proliferation of adult ECs, and prevents athero-progression[6,40,41]. Notably, endothelial Dicer suppresses EC proliferation and Notch1 signaling pathway at predilection sites of atherosclerosis[12], and miR-103 targeting of lncWDR59 reduced EC proliferation by inhibiting Notch1 activation. Therefore, these findings support the hypothesis that miRNA silencing of lncRNAs, e.g., lncWDR59 silencing by miR-103, may promote endothelial maladaptation and atherosclerosis. Notably, lncWDR59 promoted Notch1 activation by binding to Numb, which mediates NICD degradation[28], therefore impeding Numb interaction with Notch1. Therefore, disturbed flow promotes Notch1 activation not only via miR-126–5p-mediated suppression of Notch1-inhibitor Dlk1[6], but also through lncWDR59 targeting of Notch1-inhibitor Numb. Taken together, down-regulation of miR-126–5p and upregulation of miR-103 may both contribute to EC maladaptation by inhibiting Notch1 activity.

Proliferating cells are vulnerable to mitotic aberrations and DNA damage, e.g., by oxidative stress[29], due to the accumulation of damaged DNA in MN, which can lead to mitotic catastrophe[31,32,42,43]. Although MN frequency was increased in proliferating ECs, lncWDR59 protected proliferating ECs from micronucleic DNA damage accumulation via β-catenin signaling, activated through Notch1-induced Sox17 expression. Accordingly, β-catenin can promote cell cycle checks and DNA damage repair processes in response to DNA damage in proliferating cells[44]. Moreover, the activation of β-catenin generated a negative feedback loop that limited Notch1 and protected ECs from aberrant proliferation. Hence, lncWDR59 plays a dual role in EC maladaptation by promoting EC proliferation through Notch1 activation and protecting ECs during proliferation from micronucleic DNA damage accumulation through β-catenin activation. Decreasing the effect of lncWDR59 by hyperlipidemia- and oxLDL-mediated upregulation of miR-103 increased the vulnerability of proliferating ECs to mitotic aberrations, which may limit EC regeneration and promote atherosclerosis.

Targeting of lncRNAs by miRNAs may represent a new mechanism by which ECs are reprogrammed toward a maladapted phenotype under disturbed flow.

## Methods

**In vivo TSB treatment and staining**. Following 8 weeks of HFD feeding, *Apoe*^−/− mice (The Jackson Laboratory, Bar Harbor, ME, USA) were randomized to the different experimental groups and tail vein injected once weekly for four consecutive weeks with lncWDR59 LNA-TSB or control LNA oligonucleotides (0.5 mg per 20 g; miRCURY LNA Target Site Blocker, in vivo use; Exiqon). During the injection period, mice were fed with a HFD comprising 21% crude fat, 0.15% cholesterol, and 19.5% protein. Tissues were harvested 1 week after the last injection. Paxgene-fixed/paraffin-embedded aortic roots were stained with Elastic van Gieson stain. A bright-field microscope (Leica DM6000B; Leica Microsystems) connected to a CCD camera (Leica DFC365) was used to obtain the images. The lesion area was quantified with ImageJ. Immunostaining of Ki67, γH2AX, C-X-C motif chemokine ligand 1 (Cxcl1), and von Willebrand factor (vWF) was also performed. The positive area or the number of positive cells was normalized to the lesion area or to the total number of vWF+ cells, respectively, using ImageJ. The analysis of the staining was performed in a blinded manner.

All animal experiments were reviewed and approved by the local authorities (State Agency for Nature, Environment and Consumer Protection of North Rhine-Westphalia and District Government of Upper Bavaria) in accordance with the German animal protection laws.

**RNA fluorescence in situ hybridization (FISH) on human and murine tissues**. RNA FISH assay was performed using the Affymetrix protocol (ViewRNA Cell Plus Assay, Affymetrix) with some modifications. Briefly, after DNase digestion with RNase-free DNase (Roche Diagnostics, 10776785001), tissues were incubated with custom probe oligonucleotides specific for miR-103 and lncWDR59 designed and synthesized by Affymetrix as Type 1 and 6, respectively, following the

manufacturer instructions. At the end of the hybridization, sections were blocked for 1 h with 1% bovine serum albumin blocking solution, then incubated with anti-vWF antibody.

**En face immunostaining of murine tissues for micronuclei analysis**. To study MN formation and DNA damage accumulation at predilection and non-predilection sites, aortic arches and thoracic aortas from 12-week HFD- or chow diet-fed *Apoe*^−/− mice were en face prepared and stained for CD31 and γH2AX. Cell nuclei were counterstained with 4′,6-diamidino-2-phenylindole (DAPI). The stained tissues were visualized using a Leica DM6000B fluorescent microscope. To detect the fluorescent signal in en face prepared tissues, z stacks of two-dimensional images were recorded, and deconvolution was performed using a mathematical algorithm to remove out-of-focus information (AF6000 3D deconvolution software module, Leica Microsystem). The 3D rendering was performed using the LAS X 3D algorithm module (Leica) to consider the staining and MN within CD31+ cells. For quantification, en face murine aortas were entirely quantified. Data regarding MN formation were expressed as number of CD31+ cells containing MN on total number of CD31+ cells, in percentage. Data regarding γH2AX immunostaining in the MN were expressed as number of CD31+ cells showing γH2AX+ MN normalized on total number of micronucleated CD31+ cells (previously normalized on total number of cells), expressed in percentage. The analysis of the staining was performed in a blinded manner.

**Laser-capture microdissection**. Root sections were collected on ultraviolet-sterilized and RNase-free polyester-membrane 0.9 μm FrameSlides (Leica). ECs and plaques were collected using a laser microdissection system (LMD7000, Leica) in RNase-free tubes. RNA was isolated with the PAXgene RNA MinElute kit (Qiagen) and followed by pre-amplification and reverse transcription with Ovation PicoSL WTA System V2 (NuGEN) following manufacturer's instructions.

**Cell culture and transfection**. Primary MAoECs (passage 3; PELOBiotech GmbH, Planneg, Germany) and primary HAoECs (passages 2, 3; Promocell, Heidelberg, Germany) were cultured using endothelial cell complete growth medium (Promocell) containing gentamicin (0.05 mg ml^−1; ThermoFisher).

To grow the cells under different shear stresses, MAoECs were cultured in collagen-coated perfusion chambers (μ-Slides VI^0.4, ibidi GmbH, Martinsried, Germany) and exposed to high shear stress (10 dyne cm^−2) or low shear stress (5 dyne cm^−2) for 48 h generated by the perfusion with EC complete medium (ibidi Pump System, ibidi GmbH) containing 5-ethynyl-2'-deoxyuridine (EdU, 10 μM final conc., Click-iT® EdU Alexa Fluor® 488 Imaging Kit, Life Technologies) and TSBs (TSBs in vivo ready) or control LNAs (both 50 nM final conc., Exiqon).

MAoECs were transfected with antisense oligonucleotides to block the interaction between miR-103 and lncWDR59 (TSB; 50 nM, miRCURY LNA™ microRNA Target Site Blockers; Exiqon) or scrambled controls. To inhibit lncWDR59 function, murine and human ECs were transfected with lncWDR59 GapmeRs (50 nM, LNA™ GapmeRs; Exiqon) that strongly induce the degradation of the lncWDR59 transcript in the nucleus of the ECs.

**MicroRNA target identification and quantification system (MirTrap)**. MAoECs were co-transfected with miR-103-mimics, let-7b mimics, or scrambled controls, and pMirTrap Vector using the Xfect™ MicroRNA Transfection Reagent in combination with Xfect Polymer for 24 h (all from Clontech). The pMirTrap Vector expressed a DYKDDDDK-tagged GW182 protein, member of the active RISC complex, which enabled locking of the miRNA/mRNA complex into the RISC[45]. Cell lysates were collected and separated in two parts: one was used as input RNA and extracted using the NucleoSpin RNA XS Kit (Macherey-Nagel GmbH & Co. KG), the other fraction was incubated with anti-DYKDDDDK-conjugated magnetic beads. IP and subsequent RNA isolation was performed using the NucleoSpin RNA XS Kit. Reverse transcription of input and IP samples were performed using a high-capacity complementary DNA (cDNA) reverse transcription kit (Life Technologies), followed by the amplification with gene-specific primers (Supplementary Table 1) and SYBR Green PCR Master Mix (Thermo Scientific). The fold enrichment was calculated according to manufacturer's instructions.

**Human carotid lesion samples**. Human atherosclerotic lesions were collected during carotid endarterectomy. One part was fixed in 4% paraformaldehyde while another part was immediately stored in RNAlater for RNA isolation and qPCR. The study protocol for the collection of human atherosclerotic lesions was first approved by Ethics Committee of the Medical Faculty of RWTH Aachen University. The written informed consent was obtained from all participating patients. Background information regarding the patients were reported in the Supplementary Table 4. Classification of the type of the lesion was made according to the necrotic core area using a threshold 30%[46]. Compute nonparametric Spearman's rho correlation analysis was used to investigate the effect of athero-progression on *hsa-lncWDR59*, *SOX17*, and *CDKN2A* expression.

**Statistical analysis**. All analyses were performed in a blinded way. Power calculation with StatMate (GraphPad Software) was performed to calculate the number of replicates needed for each experiment. The data represent the mean ± SEM and were compared using unpaired *t*-test, one-way analysis of variance (ANOVA), or two-way ANOVA (Prism, GraphPad Software), two-sided, unless stated otherwise. A *P* value of <0.05 was considered significant.

Additional methods are available in the Supplementary Information File (see Supplementary Methods section).

**Data availability**. All relevant data are included in the manuscript and available from the authors upon reasonable request. MiRNA expression profile from microarray is available online (https://doi.org/10.1038/ncomms10521) and in the Gene Expression Omnibus database (http://www.ncbi.nlm.nih.gov/geo/) under accession numbers GSE53433, GSE53435, and GSE114805.

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

## Acknowledgements

This work was supported by the Deutsche Forschungsgemeinschaft (DFG CRC1123; project B4). We thank Lourdes Ruiz-Heinrich for her technical assistance and Dr. Donato Santovito for his help with the immunoprecipitation.

## Author contributions

L.N. and A.S. designed the study and wrote the manuscript. L.N. carried out all experiments in vivo and in vitro and analyzed the data. C.G. helped in performing qPCRs. G.C. analyzed the RNA-seq data. Y.W. performed the analysis of human atherosclerotic lesions and the LysCre-Dicer array for lncRNAs expression. M.Z. performed monocyte isolation and helped with mice experiments. A.D.F. helped with

micronuclei analysis. P.H. contributed to EC-Dicer mice experiments. R.Z. contributed to the RNA-seq analysis. All authors discussed the results and commented on the manuscript.

## Additional information

**Competing interests:** The authors declare no competing interests.

