## [Peer Review File · Nature Communications]

Reviewers' comments:

Reviewer #1 (Remarks to the Author):

The authors have described a novel mechanism contributing to atherogenesis and arterial inflammation under the regulation of miR-103 and a non-coding transcript entitled lncWDR59. Overall the study focuses on endothelial injury and damage, disregarding other cellular subsets with obvious importance to atherosclerotic lesion formation. The manuscript is overall well written and easy to comprehend, the results are convincing, and the conclusions drawn from the investigation are clearly presented and justified. I have some comments that I believe should be experimentally addressed. Please see below:

- 1) As briefly mentioned above, the manuscript focuses on endothelial function and response to atherogenic stimuli. How about other cell subsets in lesions of importance during athero-progression? Are miR-103/lncWDR59 exclusively expressed in ECs? What about its expression in SMCs and macrophages/monocytes (only briefly mentioned upon LCM in murine lesions) – in particular in response to oxLDL treatment, which stimulates foam cell formation and necrotic core expansion in SMCs and macs? Do circulating cells, adhering to ECs under disturbed flow conditions, express miR-103 and lncWDR59?
- 2) The authors state that EC proliferation is regulated by miR-103/lncWDR59 dysfunction, but not how this affects SMC proliferation. SMC proliferation and KLF4-mediated transformation are important mechanisms of lesion progression. Again, the authors should broaden their investigation to other cell subsets of importance to plaque progression.
- 3) The authors connect miR-103/lncWDR59 with flow alterations and its effect on EC viability. However, experiments using laminar vs. turbulent flow are largely lacking. The authors should evaluate miR/lncRNA function upon modulation under different flow conditions.
- 4) The sample size leading to the data presented in figure 1a is extremely small. Is it really only n=2? Drawing conclusions from this appears rather speculative and challenging. The authors should at least measure their top hits by qRT-PCR in a larger sample size, to confirm the importance of their initial finding of the profiling study.
- 5) In addition to the in vitro modulation of miR-103 and lncWDR59, is there any detectable phenotype when modulating miR-103 (using anti-miRs) or lncWDR59 (using gapmers) in atherogenic mouse models (ApoE^{-/-} or LDLR^{-/-}) that has an effect on EC function/inflammation and lesion formation (necrotic core size, fibrous cap thickness)?
- 6) The authors should consider to perform in situ hybridization to evaluate localization of miR-103 and lncWDR59 in murine (EC Dicer flox vs. WT under HF diet conditions) and more importantly in human atherosclerotic lesions (plaques vs. controls). In particular the part on human plaques appears rather preliminary, but is in my opinion extremely important to correlate/translate the finding from the murine model to advanced human disease. The study would largely benefit from an expansion using the human plaque material. Also, background information on what type of lesions (stable, vulnerable, ruptured, eroded) and from what type of patients (age, gender, risk factors) are lacking. This has to be provided into the Supplement (unless I missed it).

Minor comment:

In the introduction section, 2nd paragraph should be KLF 4 – not Kruppel like factors four.

Reviewer #2 (Remarks to the Author):

In the manuscript entitled “miR-103 promotes endothelial maladaptation by targeting lncWDR59” authors showed that miR-103 is upregulated in response to hyperlipidemia in endothelial cells and limits endothelial proliferation by targeting long noncoding RNA lncWDR59. This is a novel lncRNA that authors identified by microarray analysis of mouse atherosclerotic arteries and RNA sequencing of murine aortic endothelial cells. Authors nicely showed that in the absence of miR-103 lncWDR59 directly interacts with the inhibitor of notch1, Numb, and thereby promotes Notch1

activation. They also showed that lncWDR59 enhances the activity of β -catenin through increased expression of Sox17 following Notch1 activation. This increased β -catenin activity limits aberrant proliferation and micronucleic DNA damage accumulation in proliferating endothelial cells. Overall, this is an interesting and novel study that provides mechanistic insights on maladaptive phenotype of vascular endothelial cells during atherosclerosis and in general on endothelial reprogramming in response to stress. However there are some questions and comments that can improve the manuscript:

Questions:

-I'm curious why authors selected miR-103 and lncWDR59 to further study and not let-7b and lncRNA leonardo? could authors comment on that and describe why they decided to focus on the role of miR-103/lncWDR59 interaction on endothelial cell function?

- could authors speculate if there could be any preference for miR-103 binding to lncWDR59 or klf4? which interaction is more relevant for endothelial maladaptive phenotype?

comments:

for manuscript's main file:

-line 208 should be description of Fig.4e but it is written 4d.

-line 217: it would be better to include how long mAoEC were treated with oxLDL.

-line 222-224: it would be better to include in the text the genotype of mouse that has been used for this experiment (ApoE^{-/-}?)

for supplementary material and figures:

-line 17: EC-Dicer flow (EC-Dicer flox)

-In supplementary figure 4 C-D (line 421-422) it is better to include the mouse genotype that has been used, according to the text it should be ApoE^{-/-} but it is better to include it in the figure legend too.

-Typo in supplementary figure 5E: transversal section (transversal section)

Reviewers' comments:

Reviewer #1:

The authors have described a novel mechanism contributing to atherogenesis and arterial inflammation under the regulation of miR-103 and a non-coding transcript entitled IncWDR59. Overall the study focuses on endothelial injury and damage, disregarding other cellular subsets with obvious importance to atherosclerotic lesion formation. The manuscript is overall well written and easy to comprehend, the results are convincing, and the conclusions drawn from the investigation are clearly presented and justified. I have some comments that I believe should be experimentally addressed. Please see below:

1) As briefly mentioned above, the manuscript focuses on endothelial function and response to atherogenic stimuli. How about other cell subsets in lesions of importance during athero-progression? Are miR-103/IncWDR59 exclusively expressed in ECs? What about its expression in SMCs and macrophages/monocytes (only briefly mentioned upon LCM in murine lesions) – in particular in response to oxLDL treatment, which stimulates foam cell formation and necrotic core expansion in SMCs and macs? Do circulating cells, adhering to ECs under disturbed flow conditions, express miR-103 and IncWDR59?

First, we would like to thank the reviewer for appreciating our work and for the very constructive comments. We fully concur with the question of the reviewer regarding the expression of miR-103 and IncWDR59 in other cell types involved in atherosclerosis. Of note, miR-103 has been demonstrated to be expressed in various cell types, such as macrophages (Nazari-Jahantigh M. et al., JCI 2012), intestinal cells (Liao Y. and Lönnnerdal B., PloS One 2010), , preadipocytes (Zhang Z et al., J Lipid Res 2018) and kidney, prostate and breast tumor cells (Zheng J. et al., Int J Mol Sci 2017; Xiong B et al., Xue D. et al., Tumor Biol 2016; Biomed Pharmacother 2017). Moreover, miR-103 is highly expressed in ECs from murine and human atherosclerotic plaques (ref #12). We have now additionally analyzed the expression of miR-103 in other cell types of importance during athero-progression, such as aortic smooth muscle cells (MAoSMCs) and macrophages (BMDM). Our data indicated that miR-103 was slightly lower expressed in MAoSMCs than in MAoECs and BMDM (revised Supplementary Figure 4A, page 16-17). Therefore, miR-103 is not exclusively expressed in ECs. In addition, as requested by the reviewer, we have performed novel experiments to study the effect of oxLDL on miR-103 expression in macrophages nor MAoSMCs. In contrast to ECs, oxLDL treatment did not affect miR-103 expression in macrophages and SMCs (novel Supplementary Figure 4A. Page 3, lines36-37 in the revised version of the manuscript).

In addition to its expression in MAoECs and BMDMs, we have now investigated the expression level of IncWDR59 in MAoSMCs. We found that, in contrast to BMDMs, IncWDR59 was expressed at a similar level in SMCs as in MAoECs (novel Supplementary Figure 4A). Moreover, similar to miR-103, IncWDR59 expression was not altered in macrophages or MAoSMCs after oxLDL treatment (novel Supplementary Figure 4A. Page 3, lines 36-37 , in the revised version of the manuscript). Thus, these findings show that oxLDL regulates miR-103 and IncWDR59 expression primarily in ECs, indicating that the interaction between miR-103 and IncWDR59 in response to oxLDL is cell type specific.

As requested by the reviewer, we have now also analyzed miR-103 and IncWDR59 expression in circulating murine monocytes from Apoe^{-/-} mice. We found that both miR-103 and IncWDR59 expression were detectable in monocytes (Table I).

Expression in circulating monocytes (n = 4 per group)	C _t	sno135/b2m C _t
miR-103	27.38±0.42	26.52±1.20
IncWDR59	31.35±0.57	18.82±1.54

Table 1: MiR-103 and IncWDR59 expression in circulating monocytes

Moreover, we have now studied the effect of TSB treatment on the IncWDR59 expression in blood cells of HFD-fed Apoe^{-/-} mice. The TSB treatment did not significantly alter the expression of IncWDR59 in the blood cells compared with control-LNA treated mice (Figure I).

These data indicate that miR-103 targets IncWDR59, although both are not exclusively expressed in ECs, primarily in ECs. In line with this conclusion, conditional Dicer knockout in myeloid cells or SMCs, which decreases miR-103 expression in vascular lesions (Nazari-Jahantigh M. et al., JCI 2012 and Sadegh MK. Et al., PloS One 2012), did not alter IncWDR59 expression (Suppl. Table 2, Zahedi et al. personal communication).

2) The authors state that EC proliferation is regulated by miR-103/IncWDR59 dysfunction, but not how this affects SMC proliferation. SMC proliferation and KLF4-mediated transformation are important mechanisms of lesion progression. Again, the authors should broaden their investigation to other cell subsets of importance to plaque progression.

We thank the reviewer for the constructive comment. We fully agree with the notion of the reviewer that SMC proliferation and KLF4-mediated transformation are important mechanisms for lesion progression.

As suggested by the reviewer, we have now performed additional experiments to study the effect of the interaction between miR-103 and IncWDR59 on SMC proliferation. We found that treatment with the TSBs did not affect the proliferation of SMCs *in vitro* (n = 4 per group) (page 4, line 28; revised Supplementary material, page 6, lanes 36-42, and novel Suppl. Fig. 4L). Notably, transfection of ECs with TSBs did not affect the expression of Klf4 (Figure 2h in the revised manuscript), indicating that the TSBs specifically inhibits the interaction between miR-103 and IncWDR59, but not with Klf4. Because the specificity of the TSB is likely independent of the cell type, we believe that TSBs do not affect the interaction between miR-103 and Klf4 in SMCs, and therefore do not alter KLF4-mediated transformation.

3) The authors connect miR-103/IncWDR59 with flow alterations and its effect on EC viability. However, experiments using laminar vs. turbulent flow are largely lacking. The authors should evaluate miR/IncRNA function upon modulation under different flow conditions.

We thank the reviewer for the constructive comment. To evaluate whether the effect of the interaction between miR-103 and IncWDR59 on EC proliferation is flow dependent, we have performed additional experiment using the ibidi flow system (revised Supplemental material page 7, lines 20-29). We subjected MAoECs to low (LSS) or high shear stresses (HSS) and determined the effect of TSBs on EC proliferation. The transfection efficiency of TSB was first confirmed by qPCR. MAoECs medium was supplemented with 5-ethynyl-2'-deoxyuridine (EdU) to analyze DNA synthesis in ECs by fluorescence microscopy. Analysis of EdU⁺ ECs indicated that treatment with TSB increased the proliferation of ECs subjected to LSS but did not affect the proliferation rate of ECs subjected to HSS (novel Figure 3c, Results page 4 lines 26-27). This result supports the hypothesis that interaction between miR-103 and IncWDR59 promotes EC maladaptation to disturbed flow by impairing EC regeneration.

4) The sample size leading to the data presented in figure 1a is extremely small. Is it really only n=2? Drawing conclusions from this appears rather speculative and challenging. The authors should at least measure their top hits by qRT-PCR in a larger sample size, to confirm the importance of their initial finding of the profiling study.

Data presented in Figure 1a belong to a microarray analysis from the aortae of two 12 weeks HFD-fed EC-Dicer WT and KO mice. The mRNA array data were previously published (ref#12) and the IncRNA expression data were derived from the same experiment. As suggested by the reviewer, we confirmed the upregulation of IncRNAs, determined in the microarray study, by qRT-PCR in a larger samples size (n = 4–7 mice per group) in the previous version of the manuscript (Fig. 1e, page 12-13, lines 1-27). In order to underline that top hits IncRNAs were validated in a larger sample size, we included the number of mice used per group also in the main text (Page 3, line 8-9 in the revised version of the manuscript).

5) In addition to the in vitro modulation of miR-103 and IncWDR59, is there any detectable phenotype when modulating miR-103 (using anti-miRs) or IncWDR59 (using gapmers) in atherogenic mouse models (ApoE^{-/-} or LDLR^{-/-}) that has an effect on EC function/inflammation and lesion formation (necrotic core size, fibrous cap thickness)?

We fully agree with the notion of the reviewer that it would be interesting to study the role of the interaction between miR-103 and IncWDR59 *in vivo*. However, we believe that inhibition of miR-103 by anti-miRs will affect, at least in part, also the interaction of miR-103 with other target genes, like Klf4, which is known to increase EC inflammation during atherosclerosis (ref#12). Moreover, the role of IncWDR59 in ECs is difficult to study by inhibition of IncWDR59 using Gapmers, because it is very low expressed in athero-prone regions. To avoid these constraints, we treated HFD-fed ApoE^{-/-} mice with TSBs or control oligonucleotides to investigate the role of the interaction between miR-103 and IncWDR59 in atherosclerosis (Page 9, lines 6-19 in the new Supplementary material part). This approach may also be more cell type specific, since our data do not suggest interaction between miR-

103 and lncWDR59 in other cell types related to atherosclerosis, like macrophages and MAoSMCs. Treatment of HFD-fed Apoe^{-/-} mice with lncWDR59-TSBs reduced atherosclerosis and necrotic core size compared to control-LNA treated mice, but did not affect fibrous cap thickness (Page 7, lines 14-23 and novel Figure 7 in the revised version of the manuscript). Injection of mice with TSBs significantly increased EC proliferation and reduced DNA damage accumulation in ECs at athero-prone sites (new Figure 7). Moreover, we did not observe significant differences in Cxcl1-expressing ECs, in line with our *in vitro* data (revised Supplementary Figure 6H). Therefore, we conclude that targeting of lncWDR59 by miR-103 in ECs increases advanced atherosclerosis by inhibiting EC proliferation.

6) The authors should consider to perform *in situ* hybridization to evaluate localization of miR-103 and lncWDR59 in murine (EC Dicer flox vs. WT under HF diet conditions) and more importantly in human atherosclerotic lesions (plaques vs. controls). In particular the part on human plaques appears rather preliminary, but is in my opinion extremely important to correlate/translate the finding from the murine model to advanced human disease. The study would largely benefit from an expansion using the human plaque material. Also, background information on what type of lesions (stable, vulnerable, ruptured, eroded) and from what type of patients (age, gender, risk factors) are lacking. This has to be provided into the Supplement (unless I missed it).

We thank the reviewer for this comment and, accordingly, we have now established a *in situ* hybridization protocol to simultaneously detect miRNAs and lncRNAs in combination with an immunofluorescence staining of an endothelial specific marker. We evaluated the localization of miR-103 and lncWDR59 in murine (EC-Dicer WT and knockout aortic roots) and human atherosclerotic lesions (Page 2, lines 46-48 and page 3, lines 1-11 in the new Supplementary material version) (new Figs. 2g and 8d in the revised version of the manuscript. Page 3, lines 46-48 and page 7, lines 32-35 in the new version of the manuscript). In line with our previous findings (ref #12), we found that miR-103 is strongly expressed in atherosclerotic ECs from EC-Dicer^{WT} mice, but it was barely detectable in ECs from EC-Dicer^{flox} mice (new Fig. 2g). Conversely, lncWDR59 staining was much more prominent in ECs from EC-Dicer^{flox} mice than in those from EC-Dicer^{WT} mice (new Fig. 2g). Moreover, the endothelial expression of miR-103 and lncWDR59 was high and low, respectively, in human atherosclerotic lesions compared with non-atherosclerotic segments derived from the edge of the endarterectomy specimen (Control, new Fig. 8d).

Moreover, we translated the finding from the murine model to advanced human disease by correlating the lncWDR59 expression with necrotic core formation (revised Fig. 8e), EC proliferation (revised Fig. 8e), Cdkn2a and Sox17 expression in human plaques has been shown in the previous version of our manuscript (now revised Figure 8e and Supplementary Figure 6l).

In addition, we have now included, as requested by the referee, the background information of the patients (new Suppl. Table 4 and Page 9, lines 26-28 in the new version of Supplementary material). Since we did not have the complete background information of one out of the 21 patients, we removed this patient from the analysis (revised Fig. 8e).

Minor comment:

In the introduction section, 2nd paragraph should be KLF 4 – not Kruppel like factors four.

The text was changed from Krüppel-like factor (Klf) four to Klf4.

Reviewer #2:

In the manuscript entitled "miR-103 promotes endothelial maladaptation by targeting IncWDR59" authors showed that miR-103 is upregulated in response to hyperlipidemia in endothelial cells and limits endothelial proliferation by targeting long noncoding RNA IncWDR59. This is a novel lncRNA that authors identified by microarray analysis of mouse atherosclerotic arteries and RNA sequencing of murine aortic endothelial cells. Authors nicely showed that in the absence of miR-103 IncWDR59 directly interacts with the inhibitor of Notch1, Numb, and thereby promotes Notch1 activation. They also showed that IncWDR59 enhances the activity of β -catenin through increased expression of Sox17 following Notch1 activation. This increased β -catenin activity limits aberrant proliferation and micronucleic DNA damage accumulation in proliferating endothelial cells. Overall, this is an interesting and novel study that provides mechanistic insights on maladaptive phenotype of vascular endothelial cells during atherosclerosis and in general on endothelial reprogramming in response to stress. However there are some questions and comments that can improve the manuscript:

Questions:

- I'm curious why authors selected miR-103 and IncWDR59 to further study and not let-7b and lncRNA Leonardo? could authors comment on that and describe why they decided to focus on the role of miR-103/IncWDR59 interaction on endothelial cell function?

First, we would like to thank the reviewer for appreciating our work and for the very positive comments. We are happy that the goal of our study was clear and comprehensible, and we also believe that our work could offer a new perspective on miRNA-target lncRNAs role in EC reprogramming toward a more adaptive phenotype.

We focused on miR-103 instead of let-7b, since we found the highest number of predicted binding sites for miR-103 in the sequences of the lncRNAs upregulated in EC-Dicerflox mice (page 3, line 2 in the revised version of the manuscript). Moreover, we confirmed enrichment of four out of five lncRNA transcripts with predicted miR-103 binding sites, whereas among let-7b predicted lncRNA targets only Leonardo was enriched. Most important, compared with the let-7b binding site in the Leonardo sequence, the miR-103 predicted binding site in the IncWDR59 sequence was canonical. Therefore, the interaction between miR-103 and IncWDR59 is likely more biologically relevant than the interaction between let-7b and Leonardo. Moreover, IncWDR59 was the most significantly upregulated lncRNA in EC-Dicer knockout mice during atherosclerosis (Figure 1 and Supplementary Table 2).

Therefore, we decided to investigate the role of miR-103 targeting of IncWDR59 transcript in ECs.

- could authors speculate if there could be any preference for miR-103 binding to IncWDR59 or klf4? which interaction is more relevant for endothelial maladaptive phenotype?

We would like to thank the reviewer for the very interesting question. Together with Klf2, Klf4 is among the most significantly flow regulated genes in ECs, targeted by miR-103 to promote EC inflammation (ref#12). We observed that treatment of MAoECs with TSBs that specifically block the interaction between miR-103 and lncWDR59 transcript did not increase Klf4 mRNA levels. Although we cannot exclude that binding of miR-103 on other putative mRNA targets, we may speculate that there is not a specific preference for miR-103 binding to lncWDR59 or Klf4 transcripts.

We believe that both miR-103/lncWDR59 and miR-103/Klf4 interactions are relevant for endothelial maladaptive phenotype. Indeed, ineffective EC regeneration, partly due to miR-103 interaction with lncWDR59, together with increased EC inflammatory activation at predilection sites, due to miR-103 targeting of Klf4, both contribute to EC maladaptive phenotype. Nevertheless, lack of Klf4-mediated inhibition of EC inflammation might increase reactive oxygen species and related DNA damage accumulation in ECs. Therefore, together with miR-103 targeting of Klf4, miR-103 inhibition of lncWDR59, which prevented DNA damage in proliferating ECs through activation of Wnt signaling, could play a synergic role in EC maladaptation.

Due to the interesting consideration emerged from the comment of the reviewer, we now included a brief discussion regarding the preference of miR-103 binding to lncWDR59 or Klf4 in the Discussion part of the revised manuscript (page 8, lines 12-20).

comments:

for manuscript s main file:

-line 208 should be description of Fig.4e but it is written 4d.

We modified the text accordingly, by indicating Fig.4d,e

-line 217: it would better to include how long MAoEC were treated with oxLDL.

We modified the text accordingly, by indicating that MAoECs were treated for 24 hours with two different doses of oxLDL. The information is also included in the novel revised Figure 5a legend.

-line 222-224: it would be better to include in the text the genotype of mouse that has been used for this experiment (ApoE^{-/-}?)

We modified the text accordingly, by indicating that aortae belong to ApoE^{-/-} mice.

for supplementary material and figures:

-line 17: EC-Dicer flow (EC-Dicer flox)

We modified the text accordingly, by indicating EC-Dicer^{flox}

-In supplementary figure 4 C-D (line 421-422) it is better to include the mouse genotype that has been used, according to the text it should be ApoE^{-/-} but it is better to include it in the figure legend too.

We modified the text in the figure legend accordingly, by indicating that mice used have an ApoE^{-/-} genetic background

-Typo in supplementary figure 5E: trasversal section (transversal section)

We modified the text in the figure legend accordingly, by indicating transversal section

REVIEWERS' COMMENTS:

Reviewer #1 (Remarks to the Author):

The authors have done an excellent job in addressing all of my concerns from the previous round. I have no further questions or reservations towards the manuscript.